# Instance-Adaptive Video Compression:
# Improving Neural Codecs by Training on the Test Set

**Ties van Rozendaal**    **Johann Brehmer**    **Yunfan Zhang**    **Reza Pourreza**    **Auke Wiggers**
**Taco S. Cohen**
**Qualcomm AI Research**[1]
`{ties, jbrehmer, yzhang, pourreza, auke, tacos}@qti.qualcomm.com`

**Reviewed on OpenReview:** `https://openreview.net/forum?id=akg6kdx0Pk`

## Abstract

We introduce a video compression algorithm based on instance-adaptive learning. On each video sequence to be transmitted, we finetune a pretrained compression model. The optimal parameters are transmitted to the receiver along with the latent code. By entropy-coding the parameter updates under a suitable mixture model prior, we ensure that the network parameters can be encoded efficiently. This instance-adaptive compression algorithm is agnostic about the choice of base model and has the potential to improve any neural video codec. On UVG, HEVC, and Xiph datasets, our codec improves the performance of a scale-space flow model by between 21 % and 27 % BD-rate savings, and that of a state-of-the-art B-frame model by 17 to 20 % BD-rate savings. We also demonstrate that instance-adaptive finetuning improves the robustness to domain shift. Finally, our approach reduces the capacity requirements of compression models. We show that it enables a competitive performance even after reducing the network size by 70 %.

## 1 Introduction

Neural compression methods have enormous potential to improve the efficiency of video coding. With video constituting the large majority of internet traffic, this has significant implications for the internet at large (Sandvine, 2019). State-of-the-art algorithms model each frame by warping the previous one with a neural estimate of the optical flow (Lu et al., 2019) or scale-space flow (Agustsson et al., 2020) and adding residuals modeled by another network. Both the optical flow and the residuals are compressed with variational autoencoders. Such neural codecs have recently achieved results on par with popular classical codecs (Agustsson et al., 2020; Rippel et al., 2021; Pourreza & Cohen, 2021) such as H.264 / AVC (Wiegand et al., 2003) and H.265 / HEVC (Sullivan et al., 2012). However, relatively little research has focused on their computational complexity, and matching the rate-distortion performance of H.266 / VVC (Bross et al., 2018) is still an open problem.

Neural video codecs depend critically on *generalization*: they are developed assuming that a good performance on training and validation datasets translates to a good performance at test time. However, this assumption does not always hold in practice, for example because of limited training data or imperfect optimization. Domain shift is a particularly challenging problem given the variety in video content and styles; for instance, neural video codecs trained on natural scene video data often perform poorly on animated sequences (Agustsson et al., 2020).

Instead, in this work, we customize the codec for the sequence to compress, relaxing the need for the codec to generalize. We find that this instance-adaptive compression method results in substantially improved compression performance for various neural codecs. Additionally, it allows us to drastically reduce the computational cost of the receiver-side networks.

---

[1] Qualcomm AI Research is an initiative of Qualcomm Technologies, Inc.

We achieve this customization by finetuning the autoencoder on the sequence to compress at test time, and making the finetuned network parameters available on the receiver side. After training on the test set, the parameters are compressed, quantized, and entropy-coded to the bitstream under a suitable prior, along with the latent code. Effectively, this procedure allows trading off encoding compute for better compression efficiency, similar to how standard codecs are able to improve compression performance when given additional encoding time.

The essential idea of instance-adaptive finetuning was recently proposed by van Rozendaal et al. (2021). The authors demonstrated the idea on I-frame compression, i.e. compressing a set of similar images, but did not apply it to video sequences yet. In this work, we benchmark this method on the compression of full videos, in which only periodic key frames are compressed as images and most frames are encoded relative to one or two reference frames. Our method is general and can be applied to various settings and base models. We first demonstrate it in a setting with I-frames and P-frames using a scale-space flow architecture (Agustsson et al., 2020) as base model. Next, we show its performance in a B-frame setting, using the base model proposed by Pourreza & Cohen (2021).

On the UVG-1k, HEVC class-B, and Xiph-5N datasets, our new instance-adaptive video codec yields BD-rate savings of 17 to 27 % over the respective base models, and 5 to 44 % over the popular ffmpeg (x265) implementation of the H.265 codec (FFmpeg; VideoLAN, b). In addition, instance-adaptive finetuning lends itself to a possible reduction in model size, because a smaller network may suffice to model a single instance. We show that in this framework, smaller models can still outperform most neural codecs while reducing the computational complexity of the decoder by 70 %. We demonstrate that, unlike most neural video codecs, our method can trade off encoding compute against compression performance, and that this trade-off is more effective than for standard codecs.

## 2 Related work

### 2.1 Neural video compression

The standard framework used by most neural compression codecs (either implicitly or explicitly) is that of the variational (Kingma & Welling, 2014) or compressive (Theis et al., 2017) autoencoder. An encoder (or approximate posterior) $q_\phi(z|x)$ maps a data point $x$ to a latent representation $z$. This latent is transmitted to the decoding party by means of entropy coding under a latent prior $p_\theta(z)$. The receiver can then reconstruct the sample with a decoder (or likelihood) $p_\theta(x|z)$. The encoder, prior, and decoder are neural networks, parameterized by weights $\phi$, $\theta$ as indicated by the subscripts.

These models are trained by minimizing the rate-distortion loss

$$\mathcal{L}_{\text{RD}}(\phi, \theta) = \mathbb{E}_{x \sim p(x)} \left[ \beta \underbrace{\mathbb{E}_{z \sim q_\phi(z|x)} \left[ -\log p_\theta(z) \right]}_{R_z} + \underbrace{\mathbb{E}_{z \sim q_\phi(z|x)} \left[ -\log p_\theta(x|z) \right]}_{D} \right], \tag{1}$$

which combines a distortion metric $D$ and a rate term $R_z$ that approximates the bitrate necessary to transmit the latent code $z$. Up to a constant entropy term $H[q_\phi]$, this loss equals the $\beta$-weighted VAE loss (Habibian et al., 2019; Higgins et al., 2017).

Recent research has focused on the design of efficient network architectures for neural video codecs, in particular with respect to the temporal structure. While some earlier works were based on 3D convolutions (Habibian et al., 2019; Pessoa et al., 2020), recent models use optical flow and residual modeling to exploit similarities between frames. The latter class of approaches can be divided into predictive or P-frame temporal modeling, where the model for each frame $x_t$ is conditional on the previous frame(s) (Agustsson et al., 2020; Chen et al., 2019; Liu et al., 2020; Lombardo et al., 2019; Lu et al., 2019; Rippel et al., 2019; Golinski et al., 2020; Rippel et al., 2021; Hu et al., 2020; Yang et al., 2020b), and bidirectional or B-frame modeling, where each frame is conditioned on past as well as future reference frames (Cheng et al., 2019; Choi & Bajić, 2019; Djelouah et al., 2019; Wu et al., 2018; Park & Kim, 2021; Pourreza & Cohen, 2021; Yang et al., 2020a; Lee et al., 2022). Agustsson et al. (2020) generalize optical flow to scale-space flow by adding dynamic blurring to the warping operation, improving the modeling of uncertainty and leading to state-of-the-art results.

Table 1: Overview of neural instance-adaptive data compression methods. We categorize these works by the data modality and by the adaptation of receiver-side components. The number of adapted network parameters on the decoder side is a proxy for adaptation flexibility, although flexibility is also affected by other measures such as the frequency of adaptation, or the quantization of updates.
*Ballpark estimate based on the reported network architecture.

| Type | Work | Modality | Decoder-side adaptation | |
| --- | --- | --- | --- | --- |
| | | | Method | Parameters |
| Encoder-only | Campos et al. (2019) | image | – | – |
| | Guo et al. (2020) | image | – | – |
| | Lu et al. (2020) | video | – | – |
| | Yang et al. (2020c) | image | – | – |
| Hybrid | Lam et al. (2019) | image | overfit restoration network | 0.3 M* |
| | He et al. (2020) | video | overfit restoration network | 0.2 M* |
| | Klopp et al. (2020) | video | overfit restoration network | <0.1 M* |
| | He et al. (2021) | video | overfit restoration network | 0.2 M* |
| | Klopp et al. (2021) | video | overfit super-resolution network | <0.1 M* |
| | Liu et al. (2021) | video | overfit super-resolution network | <0.1 M* |
| Limited decoder-side adaptation | Aytekin et al. (2018) | image | choose from 4 decoders | 9.6 M* |
| | Zou et al. (2020) | image | choose from 255 bias clusters | <0.1 M* |
| | Wang et al. (2021) | image | choose from 8 decoders | 0.9 M |
| Neural Implicit | Dupont et al. (2021) | image | train decoder network from scratch | <0.1 M* |
| | Strümpler et al. (2021) | image | overfit full decoder network | <0.1 M* |
| | Chen et al. (2021) | video | train decoder network from scratch | 12.5 M |
| | Zhang et al. (2022) | video | train decoder network from scratch | <0.1 M* |
| | Lee et al. (2022) | video | train decoder network from scratch | 12.4 M* |
| Full-model adaptation | van Rozendaal et al. (2021) | I-frames | overfit full decoder network | 4.2M |
| | Mikami et al. (2021) | images | overfit full decoder network | 0.2 M* |
| | This work | video | overfit full decoder network | 18.0 M |

## 2.2 Instance-adaptive compression

Rate-distortion autoencoders are trained by minimizing the RD loss in Eq. (1) over a training dataset $\mathcal{D}$. This approach relies on the assumptions that the model is not only able to fit the training data well, but also generalizes to unseen datapoints at inference time. In practice, however, finite training data, limited model capacity, optimization imperfections, and domain shift — differences between train and test distributions — can degrade the performance.

This problem can be solved by adapting video codecs to each sequence at test time. Previous work on such approaches can be roughly categorized into encoder-only finetuning, limited decoder finetuning, implicit neural codecs, and hybrid classical-neural codecs. We provide an overview of existing methods in Tbl. 1.

Works in the first category (e. g. Lu et al. (2020)) optimize the encoder parameters $\phi$ for each data point. Such an update does not have to be communicated to the receiver side. This approach alleviates the generalization problem for the encoder, but does not solve it for the decoder and prior. We will demonstrate later that encoder-only finetuning leads to a limited improvement in compression performance.

The second category of works also adapt parts of the prior and decoder for each video instance. Since these modules need to be known on the receiver side, such an update needs to be signaled in the bitstream. Depending on the implementation, this may increase the bitrate substantially. Some works (Aytekin et al., 2018; Zou et al., 2020; Wang et al., 2021) address this by only allowing limited changes to the decoder-side models, for example, choosing one out of a number of fixed decoder networks, thus limiting the potential $RD$ gains of these methods. Finally, the third category of works propose a hybrid approach, where a video is first compressed by a classical codec and a finetuned enhancement network is added to the bitstream (He et al., 2020; 2021; Klopp et al., 2020; 2021; Liu et al., 2021). Although these methods have shown great performance at low bitrates, they tend to perform less well at higher bitrates and do not benefit from developments in end-to-end neural compression.

The work of van Rozendaal et al. (2021) introduces a method that adapts the *full* model to a *single* datapoint. The key idea is that the parameter *updates* $\delta = \theta - \theta_{\mathcal{D}}$, where $\theta_{\mathcal{D}}$ are the global model parameters resulting from training on $\mathcal{D}$, can be efficiently transmitted. After discretization, the quantized updates $\bar{\delta}$ are entropy-coded under a parameter update prior $p[\bar{\delta}]$ that assigns high probability mass (and thus low transmission cost) to zero-updates $\bar{\delta} = 0$. This procedure finetunes the full model — encoder, decoder, and prior — on each data sample at test time, by minimizing the following loss:

$$\mathcal{L}_{\text{InstA}}(\phi, \delta) = \mathcal{L}_{\text{RD}}(\phi, \theta_{\mathcal{D}} + \bar{\delta}) + \beta \underbrace{(-\log p(\delta))}_{R_\theta}. \tag{2}$$

The $R_\theta$ term reflects the increased bitstream length from coding the parameter updates $\bar{\delta}$. It therefore acts as a regularization term that incentivizes the finetuning procedure to produce updates that are cheap to encode under the parameter update prior. In van Rozendaal et al. (2021), this approach was successfully demonstrated on I-frame compression, essentially compressing a set of images with similar content. Since I-frame models cannot exploit temporal redundancy in video sequences it remains an open question how performant this method is when applied to efficient base codecs for video compression. In this work, we adapt it to the compression of full video sequences, as we will describe in the following section.

Related to this technique is a class of methods called neural-implicit compression (Dupont et al., 2021; Strümpler et al., 2021; Zhang et al., 2022; Chen et al., 2021; Lee et al., 2022), where a learned function maps coordinates to a point in pixelspace. In contrast to instance-finetuning, these methods typically involve completely fitting the model on the instance without coding any latents in the bitstream. These methods have some advantages over our work as they do not suffer from data selection bias and might be easier to standardize. However, they also come with major disadvantages. The biggest issue is that for most neural implicit methods the entire bitstream needs to be received before the first frame can be decoded. This makes them impractical for longer videos and for streaming settings. In contrast, our method only requires a very small fraction of the bitstream for the model updates after which the video can be decoded following normal VAE protocol. Furthermore, neural implicit methods typically have a poor initialization and thus need to train for a long time to get up to performance. In contrast, our method benefits from the fact that even after finetuning for 0 steps performance is already as good as the baseline.

Finally, our method is also related to the field of model quantization (Nagel et al., 2021) and model compression (Kuzmin et al., 2019; Gale et al., 2019) as we quantize the model updates to a uniform grid and entropy-code them under a prior. Our approach for parameter transmission is relatively simple and might benefit from advances in this field such as non i.i.d. priors, structured sparsity (Li et al., 2020a), advanced pruning techniques (Liu et al., 2019), and non-uniform quantization (Li et al., 2020b). A key difference with such works is that we compress and quantize model *updates* instead of model *parameters* and adapting these methods to the instance-adaptive compression setting remains an interesting direction for future research.

## 3  Instance-adaptive video compression

We now introduce our instance-adaptive compression codec for full video sequences, or InstA for short. The key idea is to optimize the parameters of a rate-distortion VAE *for each video sequence to be transmitted*, and to send the relevant network parameters in a rate-efficient way to the decoder. This approach can be beneficial whenever a standard rate-distortion autoencoder would not generalize well, for instance because of limited training data or domain shift. In addition, instance adaptation allows us to use smaller models while maintaining most of the compression performance. While encoding the network parameters to the bitstream increases the length of the code, we describe a compression scheme in which this cost is negligible, especially when amortized over many frames of a video sequence.

Our approach is agnostic about the base model, and we demonstrate it on two different architectures: a scale-space flow model in the P-frame setting, and the B-EPIC model in the B-frame setting. We first describe the base models, before describing how InstA compresses and decompresses video sequences.

Table 2: Parameter counts and decoding complexity for our various scale-space-flow base models. Compression curves comparing all four models can be seen in the left panel of Fig. 7.

| Model | SSF18 | SSF8 | SSF5 | SSF3 |
|---|---|---|---|---|
| Total parameters $[10^6]$ | 28.9 | 13.8 | 8.1 | 4.9 |
| Decoder-side parameters $[10^6]$ | 18.0 | 8.4 | 5.0 | 3.1 |
| Decoder-side kMACs / pixel | 285.8 | 167.1 | 84.7 | 46.0 |
| Decoder kMACs reduction | 0 % | 43 % | 70 % | 84 % |

### 3.1 Base model for P-frame setting: Scale-space flow (SSF)

We first focus on a *P-frame setting*, which we define as having only access to the current and previous frame when decoding a frame. As a base model for this setting, we use the scale-space-flow architecture introduced by Agustsson et al. (2020). Any video sequence is first split into *groups of pictures* (GoP). The first frame in each GoP is modeled as an image without any dependency on previous frames, i.e. as an I-frame. All other frames are P-frames, modeled as

$$x_i = \text{Scale-Space-Warp}(x_{i-1}, g_i) + r_i \,, \tag{3}$$

where $x_{i-1}$ is the reconstructed previous frame, $g_i$ is the estimated scale-space flow field, $r_i$ is the estimated residual, and the scale-space warping operation performs optical-flow warping of the previous frame with a dynamic, position-dependent amount of Gaussian blur. The I-frame images, P-frame scale-space flow $g_i$, and P-frame residuals $r_i$ are compressed with separate *hyperprior* models (Ballé et al., 2018), which have a hierarchical variational autoencoder architecture.

We largely follow the architecture choices of Agustsson et al. (2020) and refer to this model as SSF18 after its number of decoder-side parameters (in millions). In Appendix B we describe the architecture in detail and highlight the differences to Agustsson et al. (2020).

### 3.2 Smaller base models for P-frame setting

As our instance-adaptive approach is based on finetuning the compression model on a low-entropy "dataset" (a single video sequence), we hypothesize that it does not require the full expressivity of the computationally complex SSF18 model. We propose three alternative scale-space flow architectures with reduced computational complexity. In Tbl. 2 we list the parameter counts and the number of multiply-accumulate (MAC) operations required for decoding a sequence, which are reduced by 43–84% compared to the SSF18 baseline. The architectures of the four SSF models are described in detail in Appendix B.

### 3.3 Base model for B-frame setting: B-EPIC

We also consider a less constrained setting, in which frames can be compressed as B-frames as well, i.e. using both a previous frame and a future frame as reference points. This flexibility allows for even more efficient compression and is used, for example, for on-demand video streaming. As a base model we choose the B-EPIC architecture (Pourreza & Cohen, 2021). Again, the video is split into GoPs. The very first frame of a video is modeled as an I-frame, while the last frame in each GoP is a P-frame, using the last frame of the preceding GoP as a reference; both I-frame and P-frames are compressed as in the SSF model.

Any other frame $x_i$ is modeled as a B-frame: it is assigned a past reference frame $x_j$ ($j < i$) and a future reference frame $x_k$ ($k > i$); an off-the-shelf Super-SloMo frame interpolator (Jiang et al., 2018) is used to interpolate between these two reference frames. The interpolated frame is then used as a basis for scale-space flow warping:

$$x_i = \text{Scale-Space-Warp}(\text{Super-SloMo}(x_j, x_k), g_i) + r_i \,. \tag{4}$$

The interpolated frame provides a more useful starting point for optical-flow warping than the previous frame, e.g. because the combination of past and future knowledge may avoid occlusion effects.

Again, I-frames, optical flow, and residuals are modeled with three separate hyperprior models and the model is trained on the RD loss in Eq. (1). We only consider a single configuration and use hyperparameters and checkpoints from Pourreza & Cohen (2021). This model has 38.5 million parameters, 23.2 million of which are on the decoder side.

### 3.4 Encoding a video sequence

Our procedure follows van Rozendaal et al. (2021) and mainly differs in the choice of hyper-parameters and application to video auto-encoder models. For completeness we shall describe the full method here. A video sequence $x$ is compressed by:

1. Finetuning the model parameters $(\theta, \phi)$ of the base model on the sequence $x$ using Eq. (2),

2. computing the latent codes $z \sim q_\phi(z|x)$,

3. parameterizing the finetuned decoder and prior parameters as updates $\delta = \theta - \theta_\mathcal{D}$,

4. quantizing latent codes $z$ as well as prior and decoder parameter updates $\delta$, and

5. compressing the quantized latents $\bar{z}$ and updates $\bar{\delta}$ with entropy coding to the bitstream.

To finetune a pretrained base compression model, we start with the global model with parameters $(\theta_\mathcal{D}, \phi_\mathcal{D})$ and minimize the rate-distortion loss given in Eq. (2) — but only over the single video sequence $x$. This modified rate-distortion loss explicitly includes the bitrate required to send model updates $\delta$ under an update prior $p(\delta)$. We compute the regularizing $R_\theta$ loss with the unquantized updates $\delta$, but use the quantized parameter updates $\theta_\mathcal{D} + \bar{\delta}$ to calculate the $D$ and $R_z$ loss terms, using a straight-through estimator (Bengio et al., 2013) in the backward pass.

Sending the updated network parameters of course adds to the length of the bitstream. This is even the case when finetuning does not lead to changed parameters ($\bar{\delta} = 0$). Given the large size of the neural model we consider, it is therefore essential to choose an update prior that assigns a large probability mass to zero-updates $\bar{\delta} = 0$. This allows the network to transmit trivial updates at a negligible rate cost while giving it the freedom to invest bit cost in non-trivial parameter updates that improve the performance substantially. We use a *spike-and-slab* prior (Johnstone & Titterington, 2009; van Rozendaal et al., 2021), a mixture model of a narrow and a wide Gaussian distribution given by

$$p(\delta) = \frac{\mathcal{N}(\delta|0, \sigma^2 \mathbb{1}) + \alpha \, \mathcal{N}(\delta|0, s^2 \mathbb{1})}{1 + \alpha}, \tag{5}$$

where the "slab" component with variance $\sigma^2$ keeps the bitrate cost for sizable updates down, and the "spike" component with the narrow standard deviation $s \ll \sigma$ ensures cheap zero-updates. The mixing weight $\alpha$ is a tunable hyperparameter.

At the beginning of the finetuning procedure, our neural model is equal to the global model. Due to the spike-slab update prior, the rate cost is only marginally increased and the compression performance is essentially equal to the global model. During finetuning, the rate-distortion performance gradually improves, giving us an anytime algorithm that we can stop prematurely to get the best compression performance within a given encoder compute budget.

After the model has converged or a compute budget has been exhausted, we use the finetuned encoder $q_\phi$ to find the latent code $z$ corresponding to the video sequence $x$. Both $z$ and the updates to prior and decoder $\delta$, which will be necessary to decode the video sequence, are quantized. To discretize the updates $\delta$, we use a fixed grid of $n$ equal-sized bins of width $t$ centered around $\delta = 0$ and clip values at the tails. The quantization of $z$ is analogous, except that we use a bin width of $t = 1$ and do not clip the values at the tails (in line with Ballé et al. (2018)).

Finally, we write the quantized updates $\bar{\delta}$ and quantized codes $\bar{z}$ to the bitstream. We use entropy coding under the quantized update prior $p(\bar{\delta})$ and finetuned prior $p_{\theta_\mathcal{D} + \bar{\delta}}(\bar{z})$, respectively. For latent codes in the tail region we use Exp-Golomb coding (Wiegand et al., 2003).

### 3.5 Decoding a video sequence

The receiver first decodes the prior and decoder updates $\bar{\delta}$ from the bitstream. Once $\bar{\theta} = \theta_{\mathcal{D}} + \bar{\delta}$ is known on the decoder side, the latents $z$ are decoded with the prior $p_{\bar{\theta}}(z)$ and the video sequence is reconstructed with the decoder $p_{\bar{\theta}}(x|z)$ following standard VAE protocol. The only overhead on the decoder side is therefore the initial decoding of $\bar{\delta}$. While this results in a small delay before the first frame can be decoded, in practice this delay is very short (below 0.2 seconds in our experiments).

## 4 Experimental setup

### 4.1 Datasets

We use sequences from five different datasets. The global models are trained on Vimeo90k (Xue et al., 2019). We evaluate on the HEVC class-B test sequences (HEVC, 2013), on the UVG-1k (Mercat et al., 2020) dataset, and on Xiph-5N (van Rozendaal et al., 2021), which entails five Xiph.org test sequences. The performance on out-of-distribution data is tested on two sequences from the animated short film Big Buck Bunny, also part of the Xiph.org collection (Xiph.org). We choose an "easy" (BBB-E) and a "hard" (BBB-H) clip of 10s each, see Appendix A for details. All of these datasets consist of videos in Full-HD resolution ($1920 \times 1080$ pixels). In addition, we evaluate on HEVC class C with a lower resolution of $832 \times 480$ pixels.

### 4.2 Global models

The scale-space flow models described in Sec. 3 are trained with the MSE training setup described in Agustsson et al. (2020), except that we use the publicly available Vimeo-90k dataset. The $D$ loss is computed with (integer) quantized latents $\bar{z}$, while for the $R_z$ loss term we use noisy samples from $U(z - \frac{1}{2}, z + \frac{1}{2})$. Following Agustsson et al. (2020), we first train for 1 million steps on $256 \times 256$ crops with a learning rate of $10^{-4}$. We then conduct the MSE "finetune" stage of the training procedure from Agustsson et al. (2020) (not to be confused with instance-adaptive finetuning) for the SSF18 model, where we train on crops of size $h \times w = 256 \times 384$ with a learning rate of $10^{-5}$. The models are trained with a GoP size of 3 frames, which means that we split the training video into chunks of 3 frames and randomly sample chunks during training. We finally evaluate the models with a GoP size of 12. Further increasing the GoP size leads to diminishing returns in rate-distortion performance, as we demonstrate in Appendix D.

For the B-EPIC model, we use the model trained by Pourreza & Cohen (2021) on Vimeo-90k. The setup is similar to that for the SSF models except that B-EPICs' more complicated GoP structure requires training with a GoP size of 4 frames. At test time we use a GoP size of 12, the frame configurations are described in Appendix B.

### 4.3 Instance-adaptive finetuning

On each instance, we finetune the models with the InstA objective in Eq. (2), using the same weight $\beta$ as used to train the corresponding global model. We finetune for up to two weeks, corresponding to an average of $300\,000$ steps.

In the P-frame scenario we use a GoP size of 3 and finetune on full-resolution frames ($1920 \times 1080$ pixels) with a batch size of 1 and a learning rate of $10^{-5}$. After finetuning, we transmit sequences with a GoP size of 12. The prior and decoder updates $\delta = \theta - \theta_{\mathcal{D}}$ are quantized and coded under the spike-and-slab network prior described in Sec. 3.

While the SSF model follows a stationary distribution over frames, the B-EPIC model has a more complex frame structure: the distance between a frame and its reference points can vary between 1 to 6 frames. For this reason, we found it beneficial to use a GoP of 12 for InstA finetuning of the B-EPIC model. Due to memory limitations, we finetune on horizontal crops of size $256 \times 1080$ pixels instead of full-resolution frames. For the low-bitrate working points use a learning rate of $10^{-5}$, in line with Pourreza & Cohen (2021). For the high-bitrate region ($\beta \leq 0.0008$), we found that we can get faster convergence by using a learning rate of $5 \cdot 10^{-5}$.

### 4.4 Baselines

As our instance-adaptive video compression method can be applied to any compression architecture, we focus on comparing to the respective base models: we compare InstA-SSF to our reimplementation of the SSF model (Agustsson et al., 2020) and InstA-B-EPIC to the B-EPIC model (Pourreza & Cohen, 2021). To better understand where the benefits of instance-adaptive finetuning come from, we consider an additional SSF18 baseline in which only the encoder is finetuned on each instance. Unlike our InstA method, encoder-only finetuning does not require sending any model updates in the bitstream. In addition, we show several neural baselines, focusing on those with the strongest published results as well as those that are most closely related to our method.

We also run the popular classical codecs H.264 (AVC) (Wiegand et al., 2003) and H.265 (HEVC) (Sullivan et al., 2012) in the ffmpeg (x264 / x265) implementation (FFmpeg; VideoLAN, a;b) as well as the (slower but more effective) HM reference implementation (HEVC) of H.265.[1] To ensure a fair comparison, in the P-frame setting we restrict all codecs to I-frames and P-frames and fix the GoP size to 12. In the B-frame experiments, we use default settings, allowing the codecs to freely determine frame types and GoP size. For ffmpeg, we compare several different encoder presets to study the trade-off between encoding time and RD performance. In Appendix C we provide the exact commands used to generate these baseline results.

### 4.5 Metrics

We evaluate the fidelity of the reconstructions through the peak signal-to-noise ratio (PSNR) in RGB space. To summarize the rate-distortion performance in a single number, we compute the Bjøntegaard Delta bitrate (BD-rate) (Bjøntegaard, 2001), the relative rate difference between two codecs averaged over a distortion range. We evaluate the BD-rate savings of all methods relative to the HM reference implementation of HEVC, using PSNR as the distortion metric, and cubic spline interpolation. BD-rate savings are computed over a certain distortion range. We choose the largest possible range for which we have results for our methods as well as the main baselines, the values of these ranges are shown in Table 6 in the Appendix.

## 5 Results

### 5.1 Compression performance

Figures 1 show the rate-distortion curves of our instance-adaptive video codec (InstA) as well as neural and traditional baselines. Both for SSF in the P-frame and B-EPIC in the B-frame setting, the instance-adaptive models clearly outperform the corresponding base models. Finetuning only the encoder — which does not require sending model updates in the bitstream — leads to a much more modest improvement over the base models.

In the P-frame setting, InstA-SSF outperforms all other neural models as well as the ffmpeg implementations of H.264 and H.265 at all studied bitrates, except for OTU (Klopp et al., 2021) and ELF-VC (Rippel et al., 2019), which hold an edge in the low-bitrate region, M-LVC (Lin et al., 2020), which is competitive at medium bitrates, as well as C2F (Hu et al., 2022), which appeared concurrently with this method and performs strongly. Within the restrictions of the P-frame setting and at medium to high bitrates, it is even competitive with the HM reference codec. In the B-frame configuration, InstA-B-EPIC outperforms all baselines from around 0.1 bits per pixel except HM.

In Fig. 2, we show the relative compression performance compared to the HM baseline. In addition to the classical codecs and our base models, we also compare to the neural baselines proposed in Lu et al. (2019; 2020); Rippel et al. (2021); Lin et al. (2020); Klopp et al. (2020); Yang et al. (2020a;c); Hu et al. (2021; 2020; 2022); Lee et al. (2022). Relative to classical codecs, almost all neural codecs perform better at higher bitrate. The exception is Klopp et al. (2020), which consists of a classical codec followed by neural enhancement.

---

[1] We use HM version 16.19.

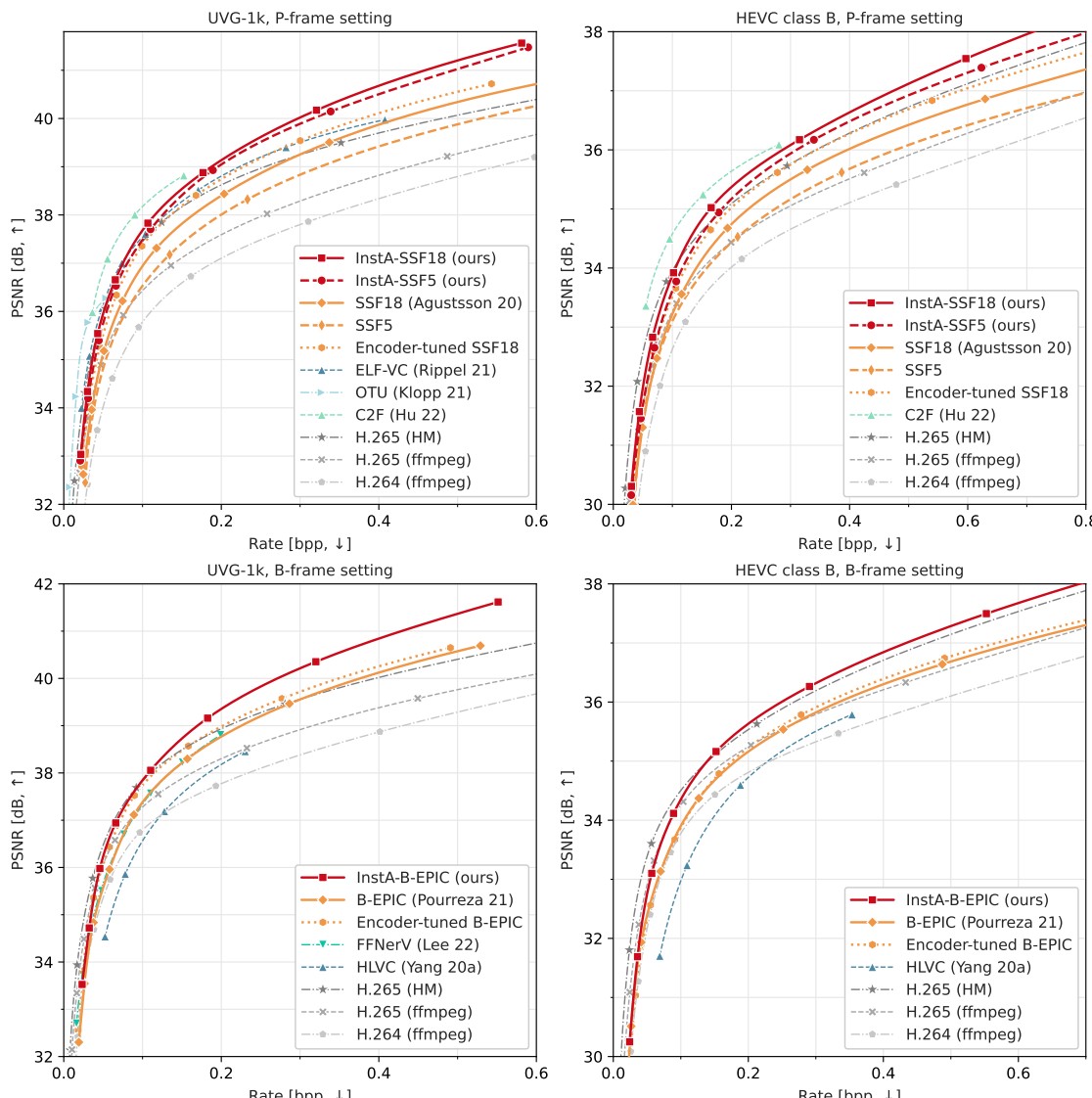

Figure 1: Rate-distortion performance of our InstA video codecs. InstA (red) leads to BD-rate savings of 17 % to 27 % over the corresponding base models (orange) and 5 % to 44 % over H.265 (ffmpeg). For all figures, higher and to the left is better.

## 5.2 Robustness to model-size reduction

While the full-size InstA-SSF18 model leads to the best RD performance in the P-frame setting, the smaller InstA-SSF5 models hold their own. Despite the 70 % reduction in decoder complexity, Insta-SSF5 outperforms most neural baselines and the ffmpeg codecs, and achieves a performance close to that of InstA-SSF18. In comparison, the gap between the SSF5 and SSF18 baselines is larger. This provides evidence for the hypothesis that instance-adaptive finetuning reduces the capacity requirements on neural compression models.

To further investigate how the performance of our method depends on decoder complexity, we train two additional models, SSF3 and SSF8, as summarized in Tbl. 2. The results are in line with the trends already shown. While the performance of the global models degrades quickly when reducing the model size, instance-adaptive finetuning can compensate for the reduced capacity and offers more "bang for the buck" in terms of (decoding-side) model complexity. For detailed results we refer the readers to Appendix D.

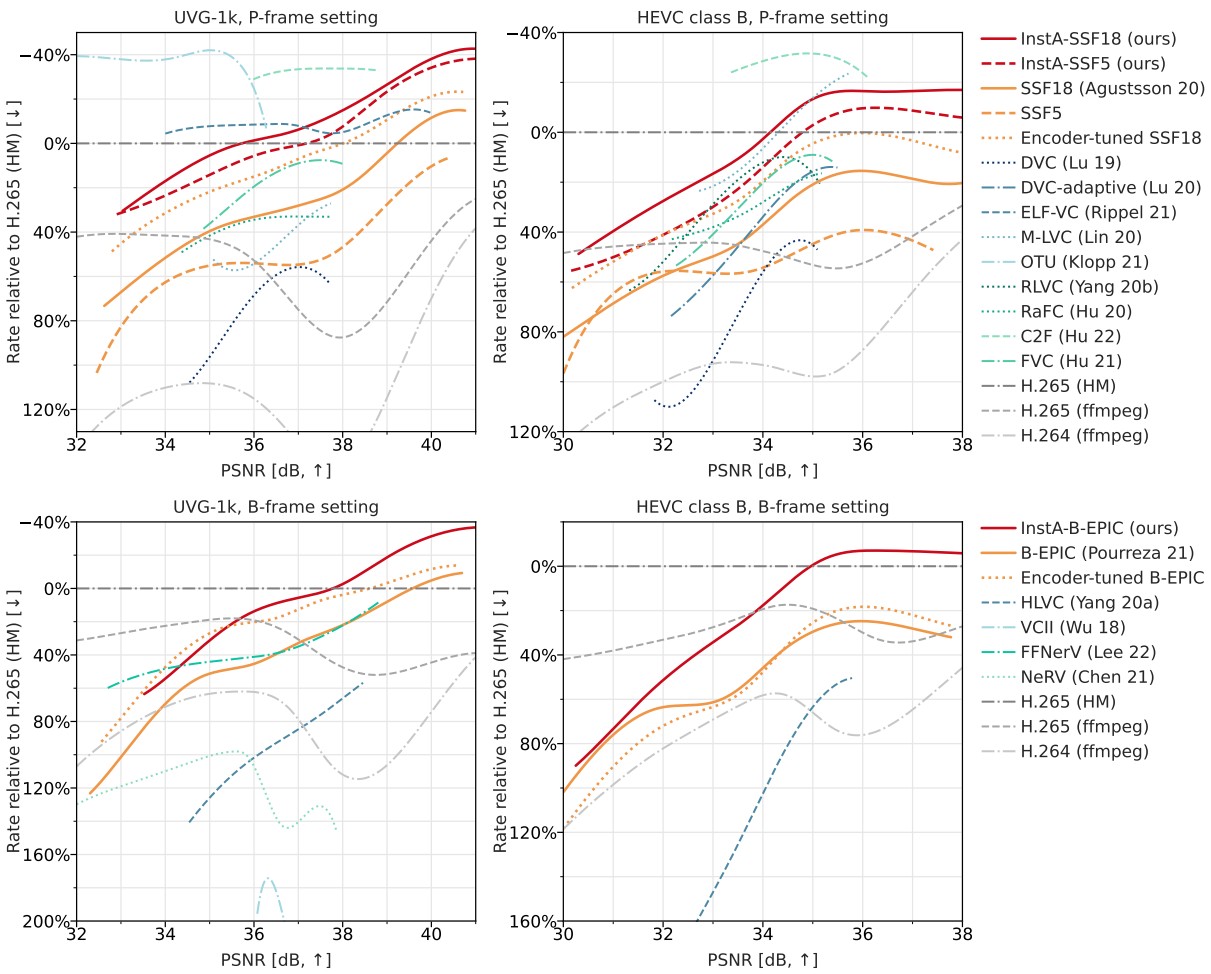

Figure 2: Bitrate of our instance-adaptive SSF models (InstA-SSF, red) relative to the HM reference implementation of H.265, as a function of the quality in dB PSNR. A relative rate of zero indicates a performance on par with HM, negative numbers (at the top) indicate fewer bits are needed, and positive numbers (at the bottom) indicate more bits are needed. This means that for all figures, higher is better. We show several neural (orange, blue, green) and traditional (grey) baselines. The interpolated curves are obtained by fitting a function in the (log) bitrate domain, similar to the way Bjøntegaard Delta-rate is computed.

Table 3: Rate-distortion performance of InstA and baselines on several datasets. We report Bjontegaard Delta-rate relative to HM, the reference implementation of H.265 (lower is better).

| Setting | Method | BD-rate relative to H.265 (HM) [↓] | | | | | |
|---|---|---|---|---|---|---|---|
| | | UVG-1k | HEVC-B | HEVC-C | Xiph-5N | BBB-E | BBB-H |
| P-frames | InstA-SSF18 (ours) | **−10.6 %** | **5.0 %** | 65.0 % | **−10.4 %** | **13.4 %** | 28.3 % |
| | InstA-SSF5 (ours) | −4.6 % | 15.4 % | 89.3 % | −7.5 % | 31.9 % | 43.2 % |
| | Encoder-finetuned SSF | 5.2 % | 21.7 % | 80.4 % | 9.1 % | 22.7 % | 48.1 % |
| | SSF18 (Agustsson et al., 2020) | 22.5 % | 38.2 % | 107.5 % | 17.5 % | 32.7 % | 54.9 % |
| | SSF5 | 43.8 % | 52.0 % | 172.4 % | 41.5 % | 84.5 % | 99.3 % |
| | H.265 (ffmpeg) | 58.8 % | 47.1 % | **42.2 %** | 29.5 % | 25.9 % | **18.7 %** |
| | H.264 (ffmpeg) | 113.8 % | 92.9 % | 73.4 % | 69.3 % | 108.4 % | 46.9 % |
| B-frames | InstA-B-EPIC (ours) | **4.6 %** | **21.9 %** | | | | |
| | Encoder-finetuned B-EPIC | 12.6 % | 48.1 % | | | | |
| | B-EPIC (Pourreza & Cohen, 2021) | 30.3 % | 47.2 % | | | | |
| | H.265 (ffmpeg) | 32.8 % | 28.8 % | | | | |
| | H.264 (ffmpeg) | 80.4 % | 74.2 % | | | | |

### 5.3 Robustness to dataset variation

Globally trained neural codecs rely on the similarity of the test instances to the training data. In comparison, we expect our finetuned models to be more robust under domain shift and to perform well across different test sets. To put this to the test, we evaluate the InstA-SSF models and P-frame baselines on different sequences, including two animated scenes from the Big Buck Bunny video. In Tbl. 3, 4, and 5 we report the compression performance as BD-rate savings relative to the HM reference codec.

We find that InstA performs strongly on all datasets. On the animated sequences, the global SSF18 baseline performs poorly relative to ffmpeg H.265, showing its susceptibility to domain shift. Our instance-adaptive models are much more robust to this shift and close the gap with classical codecs.

Instance-adaptive compression is particularly powerful for high-resolution data, where the model-rate overhead is amortized over more pixels. We test whether InstA still works at a lower resolution by evaluating on the HEVC class C test sequences, which have 81 % less pixels per frame than the Full-HD sequences. Indeed, InstA performs worse relative to the baselines than on the higher-resolution data, but it still is among the best neural codecs.

To further quantify our robustness to test data characteristics, we investigate the effect of the video framerate on the model performance. By studying the performance of global and instance-adaptive SSF models on temporally subsampled videos, we confirm that instance-adaptive model is more robust to this variation as well. In Appendix D we study the performance of global and instance-adaptive SSF models on temporally subsampled videos and again find that the instance-adaptive model is more robust to variations than the global SSF model. While these ablations do not cover all realistic variations, they serve as examples of dataset characteristics that a global neural codec is likely sensitive to.

### 5.4 Perceptual quality

We illustrate the performance on out-of-distribution data in Fig. 4, where we show original and compressed versions of two frames from the Big Buck Bunny video. Both traditional codecs and the SSF18 baseline lose details and introduce artifacts at the chosen low-bitrate setting. While the InstA-SSF18 compressions still suffer from some blurriness, artifacts are less pronounced than in the global model.

In the supplementary materials, we attach the corresponding reconstructed video sequences. To highlight the differences between the codecs, we again use a very low bitrate for all codecs (even though our InstA method is strongest at higher bitrates).

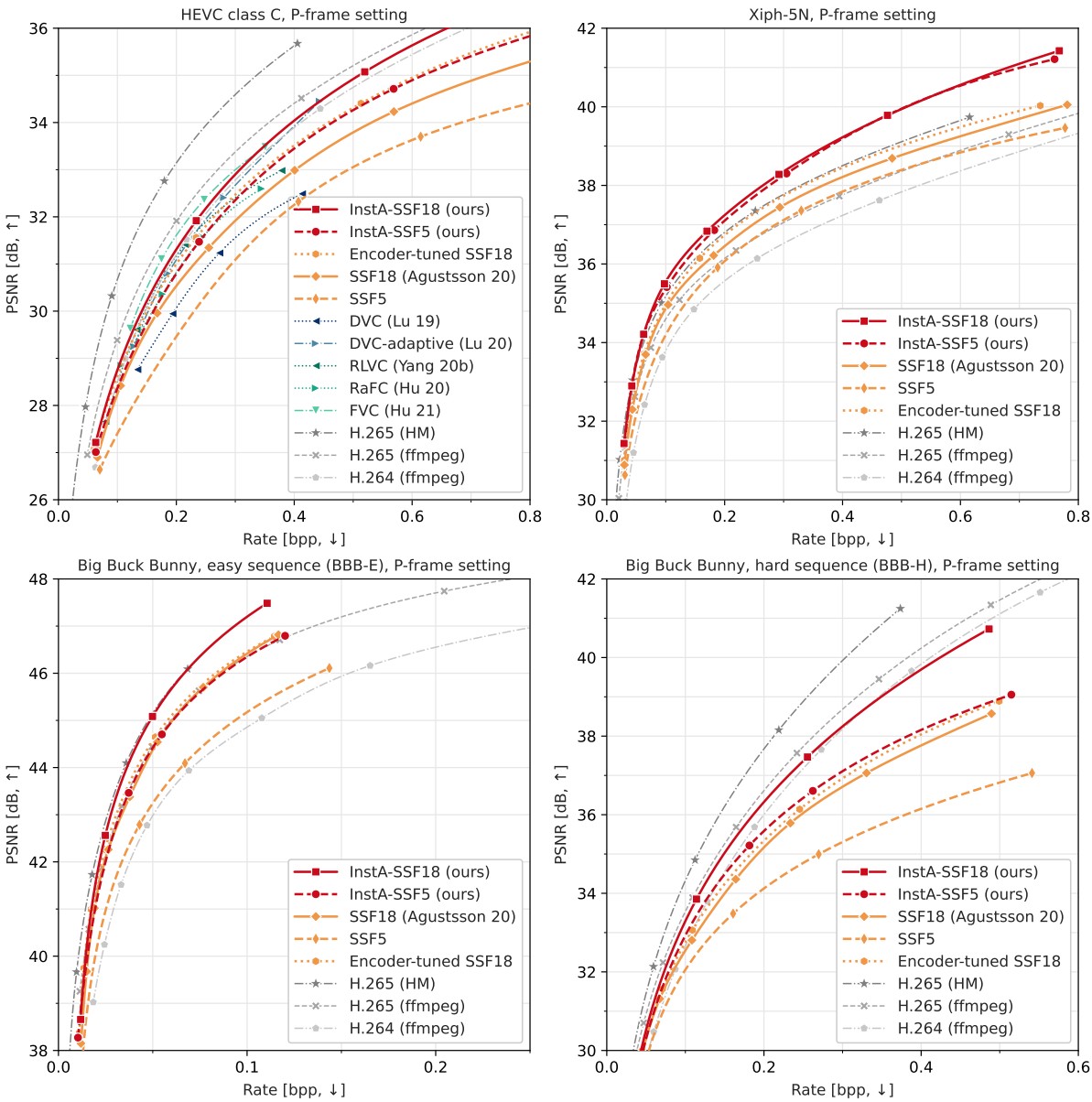

Figure 3: Rate-distortion performance of our instance-adaptive SSF models (InstA-SSF, red) compared to neural (orange) and traditional (grey) baselines. We show results on class C of the HEVC test sequences (top left), five Xiph.org test sequences (top right), and the "easy" (left) and "hard" (right) sequences from the Big Buck Bunny video (bottom). For all figures, higher and to the left is better.

Table 4: BD-rate of several codecs (rows) compared to different reference codecs (columns) in the P-frame setting. Lower numbers are better, the best results on each dataset are shown in bold.

| Dataset | Method | BD-rate relative to. . . | | | |
|---------|--------|------------|------------|------------|------|
| | | H.265 (HM) | H.265 (ffmpeg) | H.264 (ffmpeg) | SSF18 |
| UVG | InstA-SSF18 (ours) | $-\mathbf{6.7}\%$ | $-\mathbf{41.9}\%$ | $-\mathbf{57.6}\%$ | $-\mathbf{26.7}\%$ |
| | InstA-SSF5 (ours) | $-0.3\%$ | $-38.0\%$ | $-54.7\%$ | $-21.7\%$ |
| | SSF18 | $27.3\%$ | $-20.8\%$ | $-42.1\%$ | |
| | Encoder-finetuned SSF18 | $8.8\%$ | $-32.3\%$ | $-50.5\%$ | $-14.5\%$ |
| HEVC-B | InstA-SSF18 (ours) | $\mathbf{5.0}\%$ | $-\mathbf{28.6}\%$ | $-\mathbf{45.5}\%$ | $-\mathbf{24.0}\%$ |
| | InstA-SSF5 (ours) | $15.4\%$ | $-21.6\%$ | $-40.2\%$ | $-16.5\%$ |
| | SSF18 | $38.2\%$ | $-6.1\%$ | $-28.3\%$ | |
| | Encoder-finetuned SSF18 | $21.7\%$ | $-17.3\%$ | $-36.9\%$ | $-12.0\%$ |
| HEVC-C | InstA-SSF18 (ours) | $\mathbf{65.0}\%$ | $\mathbf{16.0}\%$ | $-\mathbf{4.8}\%$ | $-\mathbf{20.5}\%$ |
| | InstA-SSF5 (ours) | $89.3\%$ | $33.1\%$ | $9.2\%$ | $-8.8\%$ |
| | SSF18 | $107.5\%$ | $45.9\%$ | $19.7\%$ | |
| | Encoder-finetuned SSF18 | $80.4\%$ | $26.9\%$ | $4.1\%$ | $-13.1\%$ |
| xiph5N | InstA-SSF18 (ours) | $-\mathbf{10.4}\%$ | $-\mathbf{30.8}\%$ | $-\mathbf{47.1}\%$ | $-\mathbf{23.8}\%$ |
| | InstA-SSF5 (ours) | $-7.5\%$ | $-28.6\%$ | $-45.4\%$ | $-21.3\%$ |
| | SSF18 | $17.5\%$ | $-9.2\%$ | $-30.6\%$ | |
| | Encoder-finetuned SSF18 | $9.1\%$ | $-15.7\%$ | $-35.6\%$ | $-7.2\%$ |
| BBB-E | InstA-SSF18 (ours) | $\mathbf{13.4}\%$ | $-\mathbf{9.9}\%$ | $-\mathbf{45.6}\%$ | $-\mathbf{14.5}\%$ |
| | InstA-SSF5 (ours) | $31.9\%$ | $4.8\%$ | $-36.7\%$ | $-0.7\%$ |
| | SSF18 | $32.7\%$ | $5.4\%$ | $-36.3\%$ | |
| | Encoder-finetuned SSF18 | $22.7\%$ | $-2.5\%$ | $-41.1\%$ | $-7.5\%$ |
| BBB-H | InstA-SSF18 (ours) | $\mathbf{28.3}\%$ | $\mathbf{8.1}\%$ | $-\mathbf{12.6}\%$ | $-\mathbf{17.1}\%$ |
| | InstA-SSF5 (ours) | $43.2\%$ | $20.6\%$ | $-2.5\%$ | $-7.5\%$ |
| | SSF18 | $54.9\%$ | $30.5\%$ | $5.4\%$ | |
| | Encoder-finetuned SSF18 | $48.1\%$ | $24.8\%$ | $0.8\%$ | $-4.4\%$ |

Table 5: BD-rate of several codecs (rows) compared to different reference codecs (columns) in the B-frame setting. Lower numbers are better, the best results on each dataset are shown in bold.

| Dataset | Method | BD-rate relative to. . . | | | |
|---------|--------|------------|------------|------------|------|
| | | H.265 (HM) | H.265 (ffmpeg) | H.264 (ffmpeg) | B-EPIC |
| UVG-1k | InstA-B-EPIC (ours) | $\mathbf{9.9}\%$ | $-\mathbf{16.5}\%$ | $-\mathbf{39.4}\%$ | $-\mathbf{18.7}\%$ |
| | B-EPIC | $35.2\%$ | $2.8\%$ | $-25.5\%$ | |
| | Encoder-finetuned B-EPIC | $15.9\%$ | $-11.9\%$ | $-36.1\%$ | $-14.3\%$ |
| HEVC-B | InstA-B-EPIC (ours) | $\mathbf{21.9}\%$ | $-\mathbf{5.4}\%$ | $-\mathbf{30.0}\%$ | $-\mathbf{17.2}\%$ |
| | B-EPIC | $47.2\%$ | $14.3\%$ | $-15.5\%$ | |
| | Encoder-finetuned B-EPIC | $48.1\%$ | $15.0\%$ | $-15.0\%$ | $0.6\%$ |

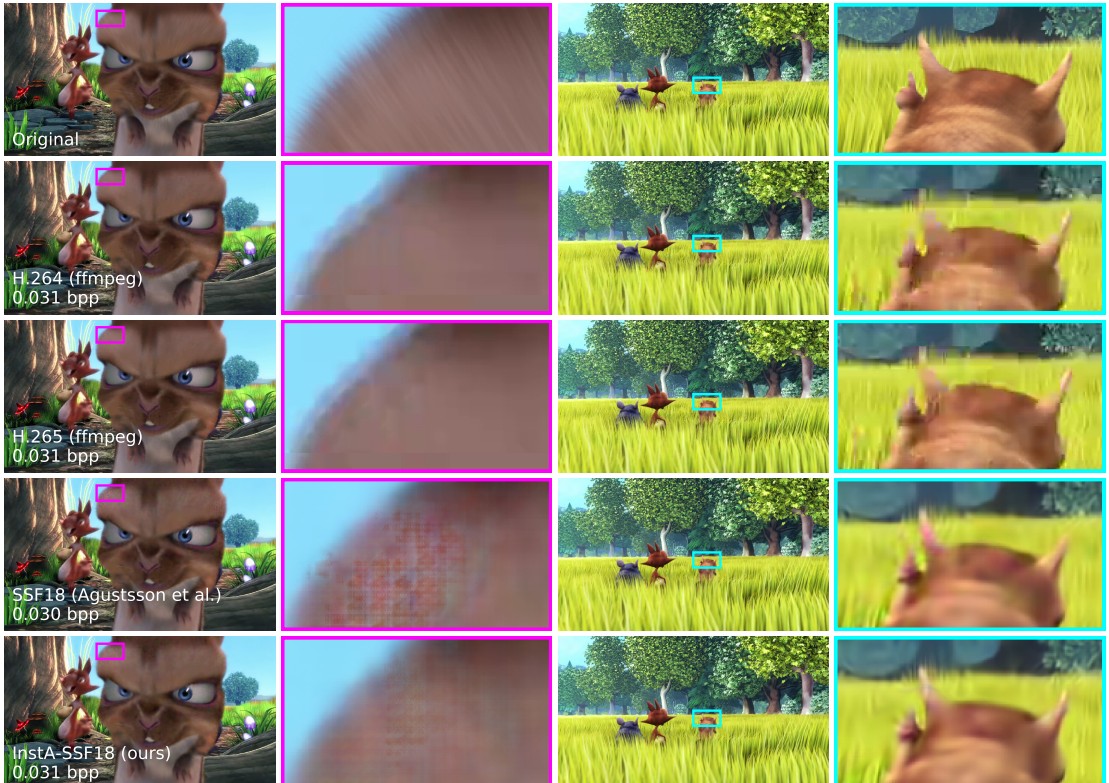

Figure 4: Original and compressed details in frames 7395 (left) and 7276 (right) of the Big Buck Bunny video (Xiph.org), which are compressed as a P-frame (left) and I-frame (right). From top to bottom, we compare the original, compressed frames with the H.264 and H.265 codecs in the ffmpeg (x264 / x265) implementation, the SSF18 baseline, and our proposed InstA-SSF18 method. We use settings with comparable, low bitrates; the achieved bitrates (over the 10s sequence) are given in the left panels. All codecs lose textural details, but our InstA-SSF18 codec reduces the artifacts visible in the SSF18 reconstructions.

## 5.5 Rate composition

It is remarkable that InstA consistently improves the rate-distortion performance of neural models despite having to send updates for millions of neural network parameters. We investigate this low rate overhead by estimating how the update prior affects the bitrate. We find that the spike-and-slab prior is indeed crucial for the low rate overhead: compared to Gaussian coding, it allows us to save around 7.5 bits per parameter (see Appendix D).

InstA learns to distribute bits on model updates and latents. The optimal trade-off generally depends on the bitrate, the size of the model and latent code, and on the test data: we expect that for strongly out-of-distribution data, more bits will be spent on model updates. We show the learned bit distribution in Fig. 9. Indeed we find that in almost all cases, less than 1% of the total rate is spent on model updates, while for the animated sequences from the Big Buck Bunny video between 1% and 3% of the total bitrate are invested in model updates.

## 5.6 Update frequency

For longer videos, the granularity of data on which the model is finetuned and the frequency with which updates are transmitted are open questions. Finetuning models on shorter scenes may improve the compression performance. Moreover, each key frame (or random access point) requires the transmission of a model update in the bitstream. On the other hand, too frequent update transmissions will increase the bitrate over-

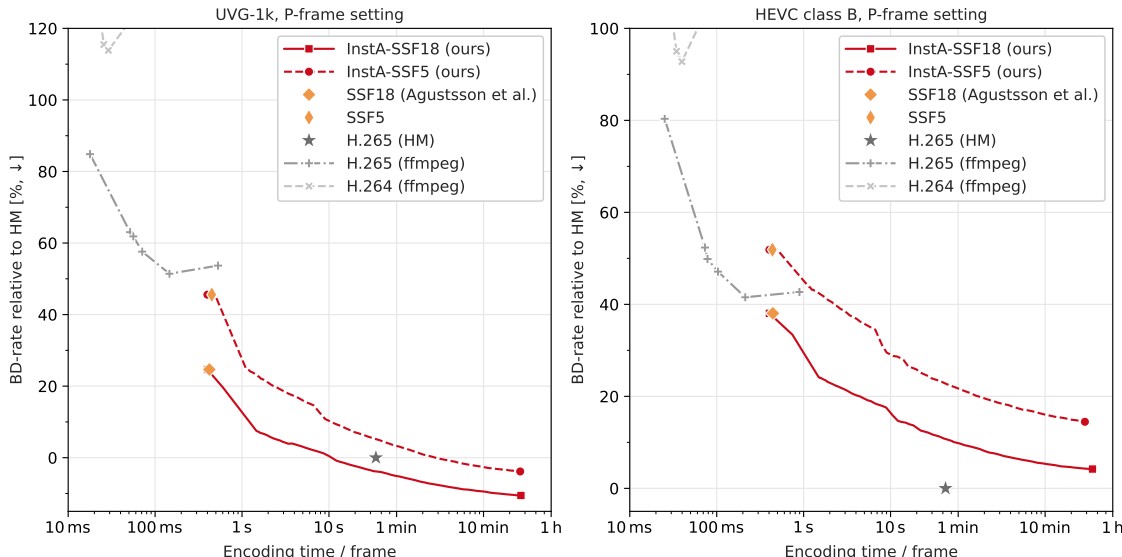

Figure 5: Trade-off between RD performance and encoder-side compute in the P-frame setting, where lower and to the left is better. We summarize the RD performance as BD-rate difference relative to the HM implementation of H.265 (lower is better). InstA-SSF (red) is compared to SSF (orange) and traditional (grey) baselines. Note that the runtime difference between neural and traditional codecs strongly depends on the available hardware, since the neural codecs require a GPU to be efficient, while the traditional codecs run on the (multi-core) CPU. Some fluctuations in the measured walltime are due to varying loads on our computational cluster. Since we can stop the instance finetuning at any time, InstA-SSF allows us to continuously trade off encoder compute for compression performance. [2]

head, limiting the improvement in rate-distortion performance from our method. The optimal compromise will depend on the application, video resolution, video content, and model complexity. In our experiments we compressed sequences in the 2.5–10 s range. Given the small overhead from the model updates, sending model updates in the bitstream once every second is certainly feasible. In Appendix D we study adaptation at smaller granularity, finding that finetuning a model for an entire video of 10 minutes provides a slightly worse performance than finetuning on scenes of 10 seconds.

## 5.7 Computational complexity

The improved compression performance of our instance-adaptive video codec comes at the price of increased encoding time. In Fig. 5, we illustrate this trade-off for the P-frame setting. We summarize the rate-distortion performance as BD-rate savings relative to HM and compare to the encoding time[2]. For our instance-adaptive methods, this time includes the finetuning in addition to the forward pass of the encoder, the quantization, and the entropy coding (for both the latents and the parameters). By choosing how long to run the finetuning, we can trade off RD performance against the encoding time. Out of the studied codecs, ours can best exploit this trade-off: encoding time and compression performance are fixed in other neural methods, while the existing handles in traditional codecs have a limited effect on performance.

We find that instance-adaptive finetuning leads to state-of-the-art RD performance even when finetuning only for a few steps. The performance gradually improves over the course of training and eventually flattens out. Care has to be taken in interpreting these results as the different codecs were designed for different hardware and our algorithms have not been optimized for performance.

Instance-adaptive video compression also adds some overhead on the decoder side, as the model parameters need to be decoded before any frame can be reconstructed. For the full-size InstA-SSF18 model and without

---

[2]We report the walltime on machines with 40-core Intel Xeon Gold 6230 CPUs with 256 GB RAM and NVIDIA Tesla V100-SXM2 GPUs with 32 GB VRAM. We only use a single GPU.

optimizing the code for performance, this initial delay is below 0.2 seconds. The neural network forward pass alone is timed with a speed of around 0.01 seconds per frame on a TeslaV100 GPU without any optimization. Both the overhead and the network forward pass are thus negligible compared to the time it takes to entropy-decode the latents $z$: in our (not optimized) implementation, this takes around 1 second per frame, both for the global SSF models and InstA-SSF. Compared to the neural baselines, InstA-SSF thus yields better performance at comparable complexity. Compared to traditional codecs, there is still a gap to bridge, and none of the neural codecs can decode at the framerate of the videos yet.

At the same time, instance-adaptive finetuning allows us to use smaller architectures while maintaining strong performance, as shown above. This potential to reduce the decoder-side compute may outweigh the small overhead from the initial update decoding.

## 6    Conclusions

We introduced InstA, a method for video compression that finetunes a pretrained neural compression model on each video sequence to be transmitted. The finetuned network parameters are compressed and transmitted along with the latent code. The method is general and can be applied to many different settings and architectures, we demonstrated it on scale-space flow models in the P-frame setting as well as a state-of-the-art B-frame architecture.

Instance-adaptive video compression relaxes the requirement of a single global model that needs to be able to generalize to any unseen test sequence. This can be beneficial when the training data is limited, when there is domain shift between training and test distributions, or when the model class is not expressive enough. While it comes at the cost of some overhead to the bitstream, we demonstrated that with a suitable compression scheme this overhead can be kept small, especially since it is amortized over the many pixels of a video sequence.

Our InstA models clearly outperform the base models in all considered settings, in several cases achieving a state-of-the-art rate-distortion performance. They outperform their corresponding global base models with BD-rate savings of between 17 % and 27 % on the UVG-1k, HEVC class-B, and Xiph-5N datasets, and the popular ffmpeg implementation of H.265 with BD-rate savings between 5 % and 44 %. By evaluating models that were trained on natural scene data on animated sequences, we demonstrate that instance-adaptive finetuning improves the robustness of neural video codecs under domain shift. Finally, instance-adaptive compression reduces the required model expressivity: even after reducing the model size by 70 %, InstA-SSF maintains a competitive performance.

The added performance improvement comes with significant added computing cost at the encoder side. However, the ability to trade off encoder-side compute for compression performance is a natural match for one-to-many transmission scenarios, in which a sequence is compressed once but transmitted and decoded often. We demonstrated that even just finetuning for seconds per frame can lead to a substantial improvement in compression performance, so instance-adaptive compression may ultimately even be beneficial in real-time applications.

### Acknowledgements

We would like to thank the whole Qualcomm AI Research team, but in particular Amir Said, Guillaume Sautiére, Yang Yang, and Yinhao Zhu for useful discussions and help with the experiments.

We are grateful to the authors and maintainers of ffmpeg (FFmpeg), Matplotlib (Hunter, 2007), Numpy (Charles R Harris et al., 2020), OpenCV (Bradski, 2000), pandas (McKinney, 2010), Python (Python core team, 2019), PyTorch (Paszke et al., 2017), scipy (SciPy contributors, 2020), and seaborn (Waskom, 2021).

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

## A Datasets

### A.1 UVG-1k

Following Wu et al. (2018); Lu et al. (2019); Golinski et al. (2020); Agustsson et al. (2020), we report results on the UVG-1k dataset, which consists of seven test video sequences totaling 3900 frames captured at 120 fps and natively available in 1080p resolution. Note that these sequences are only a subset of the full UVG dataset (Mercat et al., 2020).

### A.2 Xiph-5N

We use the Xiph-5N dataset introduced in van Rozendaal et al. (2021). It consists of the following five sequences from the Xiph.org collection: "in_to_tree", "aspen", "controlled_burn", "sunflower", and "pedestrian_area".

### A.3 Big Buck Bunny (BBB-E and BBB-H)

From the Big Buck Bunny video, which is also part of the Xiph.org media collection, we extract two subsequences of 10 s length. They are chosen based on the frame-wise RD loss achieved by a global SSF18 model (with $\beta = 0.0016$). We select the 10 s window with the lowest RD loss and the 10 s window with the highest RD loss, excluding the intro and outro of the video. These signify the "easiest" and "hardest" parts of the video to compress and we refer to them as "BBB-E" and "BBB-H", respectively. In Fig. 6 we show the frame-wise RD loss and the selected sequences.

## B Model architectures

### B.1 Full-size SSF model

We first apply instance-adaptive video compression to the scale-space flow (SSF) architecture proposed by Agustsson et al. (Agustsson et al., 2020), using our own re-implementation of the model. Groups of pictures

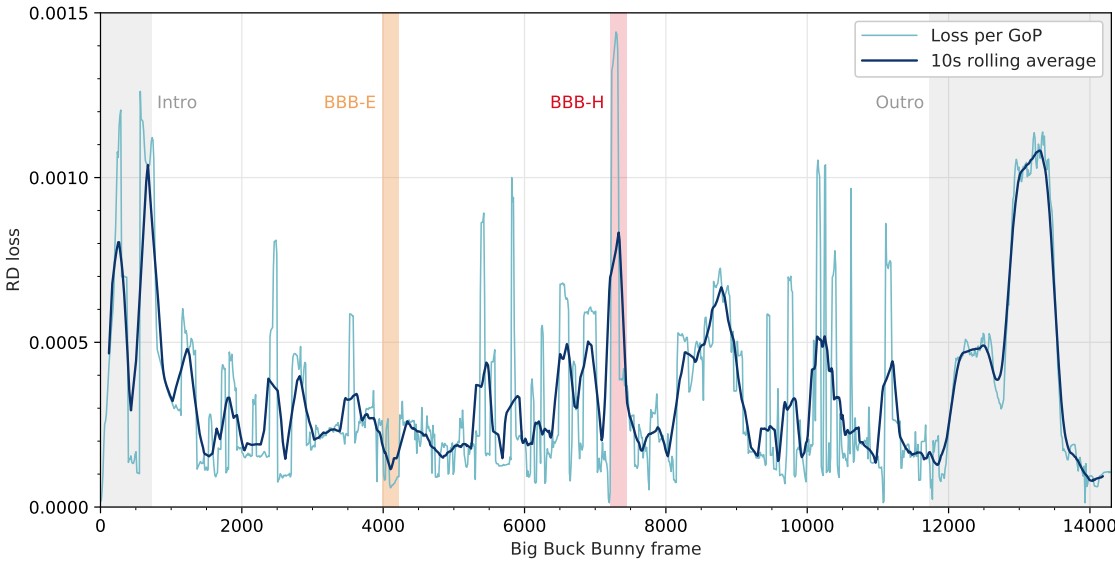

Figure 6: Temporal distribution of the RD loss of the global SSF18 model for the Big Buck Bunny video. We use the 10 s windows with the lowest and highest loss as benchmarks, excluding the intro and outro of Big Bunny. These subsequences, which we call "BBB-E" and "BBB-H" for "easy to compress" and "hard to compress", are marked in orange and red.

Table 6: PSNR ranges used to compute BD-rate savings for each dataset.

| Dataset | PSNR range |
|---|---|
| UVG-1k | $33.5\,\text{dB} \leq \text{PSNR} \leq 40.4\,\text{dB}$ |
| HEVC class B | $30.3\,\text{dB} \leq \text{PSNR} \leq 37.4\,\text{dB}$ |
| HEVC class C | $27.2\,\text{dB} \leq \text{PSNR} \leq 35.4\,\text{dB}$ |
| Xiph-5N | $31.4\,\text{dB} \leq \text{PSNR} \leq 39.5\,\text{dB}$ |
| BBB-E | $29.3\,\text{dB} \leq \text{PSNR} \leq 37.0\,\text{dB}$ |
| BBB-H | $31.4\,\text{dB} \leq \text{PSNR} \leq 39.5\,\text{dB}$ |

Table 7: Architecture differences between the original SSF model published by Agustsson et al. (2020) and our SSF18 re-implementation. Convolutional layers are indicated as "conv [kernel size] [layer type][stride] [output channels]". ↓ is used for convolutional layers, while ↑ is used for transposed convolutions. The hyperdecoder is modeled as a Gaussian $p(z_2|z_1) = \mathcal{N}(z_2|\mu(z_1), \sigma(z_1))$, where the mean and standard deviation are modeled by two separate neural networks $\mu(z_1)$ and $\sigma(z_1)$ with largely identical architectures. These networks differ in their activation functions, activations that are only used in one of the two networks are indicated in the table by their respective symbol. Note that we always clamp $\sigma(z_1)$ to the range $[0.11, \infty)$.

| Agustsson et al. (Agustsson et al., 2020) | Our reimplementation |
|---|---|
| *Hyperencoder $q(z_1|z_2)$* | |
| conv 5x5 ↓2 192 | conv 3x3 ↓1 192 |
| ReLU | ReLU |
| conv 5x5 ↓2 192 | conv 5x5 ↓2 192 |
| ReLU | ReLU |
| conv 5x5 ↓2 192 | conv 5x5 ↓2 192 |
| *Hyperdecoder $p(z_2|z_1)$* | |
| conv 5x5 ↑2 192 | conv 5x5 ↑2 192 |
| $\mu$: ReLU, $\sigma$: QReLU | ReLU |
| conv 5x5 ↑2 192 | conv 5x5 ↑2 192 |
| $\mu$: ReLU, $\sigma$: QReLU | ReLU |
| conv 5x5 ↑2 192 | conv 3x3 ↑1 192 |
| $\sigma$: QReLU | $\sigma$: ReLU |

(GoPs) consist of 12 frames; the first of them is coded as an I-frame and the remaining 11 as P-frames, each using the previous frame as reference frame. We mostly follow the setup described in Agustsson et al. (2020), but there are some minor differences. In the hyperencoder and hyperdecoder networks we use a slightly different architecture. We list the architectures in Tbl. 7. In addition, our implementation of the blur stack and of the scale-space-flow warping operation differs from the one used by Agustsson et al. (2020), but we have verified in experiments that this has a negligible effect.

We empirically validated our SSF re-implementation by comparing its performance to the original SSF (using entropy-coding). On the UVG-1k dataset, our SSF18 model performed slightly better than the results published by Agustsson et al. (2020).

## B.2    Smaller SSF models

In addition to the full-size SSF18 model, we consider three smaller scale-space flow architectures. They follow the same structure as SSF18 in Tbl. 7 but differ in the number of latent variables and convolutional channels, as shown in Tbl. 8. We refer to them as SSF8, SSF5, and SSF3 after the number of decoder-side network parameters (in millions).

## B.3    B-EPIC model

As a second base model we use the B-frame codec developed by Pourreza & Cohen (2021) (B-EPIC). We follow the hierarchical IBP configuration presented in that paper and use the model trained by the authors. During training, a GoP consists of 4 frames arranged in an IBBP pattern. At test time, we use a GoP size

Table 8: Architecture details for our scale-space-flow models. In the top, we show the number of channels for the latent and hyperlatent variables, $z$, which are the basis for the compressed code; in the bottom, we show the number of channels in the codec and hypercodec networks in the encoder and decoder. We use the same model architecture and size for I-frame, scale-space-flow and residual networks.

| Model | SSF18 | SSF8 | SSF5 | SSF3 |
|---|---|---|---|---|
| latent channels | 192 | 192 | 192 | 128 |
| hyperlatent channels | 192 | 192 | 192 | 192 |
| codec channels | 128 | 96 | 64 | 48 |
| hypercodec channels | 192 | 96 | 64 | 48 |

of 12 frames and the following reference structure (where frame 0 refers to the last frame of the previous GoP):

- Frame 12: P-frame referring to frame 0

- Frame 6: B-frame referring to frames 0 and 12

- Frame 3: B-frame referring to frames 0 and 6

- Frame 9: B-frame referring to frames 6 and 12

- Frame 1: B-frame referring to frames 0 and 3

- Frame 2: B-frame referring to frames 1 and 3

and so on for the remaining frames. The very first frame in a video is encoded as an I-frame. Encoding and decoding in this order ensures that the reference frames will always be encoded or decoded before the frames for which they are needed.

### B.4 Quantization settings

Ablation studies showed insensitivity to network prior settings in the ranges of $10 \leq \alpha \leq 1000$, $0.0005 \leq t \leq 0.005$, and $0.01 \leq \sigma \leq 0.05$. For the experiments in this paper we use a bin width $t = 0.001$, $\sigma = 0.05$, $s = t/6$, a spike-slab ratio $\alpha = 100$, and a number of quantization bins of $n = 289$.

### B.5 Entropy coding

We implement an entropy coder for the latents and model updates in the SSF and InstA-SSF models. We verified that this achieves bitrates in agreement with the probability mass functions of $p(z)$ and $p(\delta)$. For the B-EPIC and InstA-B-EPIC models, we report bitrates based on probability mass functions without actually implementing an entropy coder.

## C Classical codecs

We generate H.265 and H.264 results using version v3.4.8 of ffmpeg (FFmpeg). When comparing to neural codecs in the P-frame setting, we also choose a low-latency setting that only allows I-frames and P-frames and fix the GoP size to 12. By default, we use the encoder presets "veryslow"; to study the trade-off between encoding compute and compression performance we vary it between "veryslow" and "ultrafast". Example commands are

```
ffmpeg -pix_fmt yuv420p -s 1920x1080 \
-r 120 -i raw_video.yuv \
-c:v libx264 -preset veryslow \
-crf 23 -tune zerolatency \
```

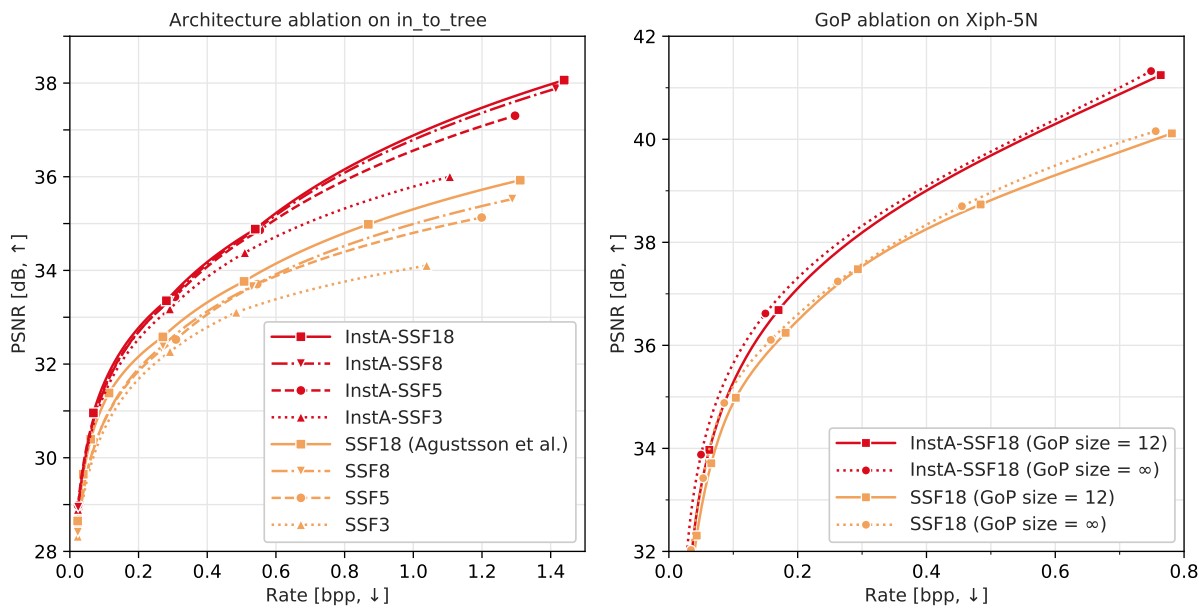

Figure 7: *Left:* Model size ablation study. We compare several global and instance-finetuned scale-space-flow models, showing the performance on the in_to_tree sequence (Xiph.org). *Right:* Effect of the GoP size at test time. We show rate-distortion curves of instance-adaptive and global SSF18 models on the HEVC Xiph-5N dataset, comparing a fixed GoP size of 12 (solid lines) to an infinite GoP size (dotted lines).

```
-x264-params \
"keyint=12:min-keyint=12:verbose=1" \
output.mkv
```

for H.264 and

```
ffmpeg -pix_fmt yuv420p -s 1920x1080 \
-r 120 -i raw_video.yuv \
-c:v libx265 -preset veryslow \
-crf 23 -tune zerolatency \
-x265-params \
"keyint=12:min-keyint=12:verbose=1" \
output.mkv
```

for H.265. We also compress the test sequences with the HM reference implementation of HEVC, using the "lowlatencyP" setting with a fixed GoP size of 12.

As baselines for the neural B-frame codecs, we also relax the GoP and frame-type restrictions for the classical codecs, using the default random-access settings with B-frames and variable GoP size.

# D Ablation studies

## D.1 Model size ablation

In the left panel of Fig. 7 we compare the compression performance of globally trained SSF18, SSF8, SSF5, and SSF3 models, as well as instance-adaptive versions InstA-SSF18, InstA-SSF8, InstA-SSF5, and InstA-SSF3. Due to the training time requirement we limit this ablation to a single video, the "in_to_tree" sequence from the Xiph.org collection. We find that the larger models lead to a better compression performance, with more pronounced difference in the high-bitrate region. SSF3 and InstA-SSF3 models perform poorly at high bitrates. We attribute this partially to the small size of the latent space compared to all other

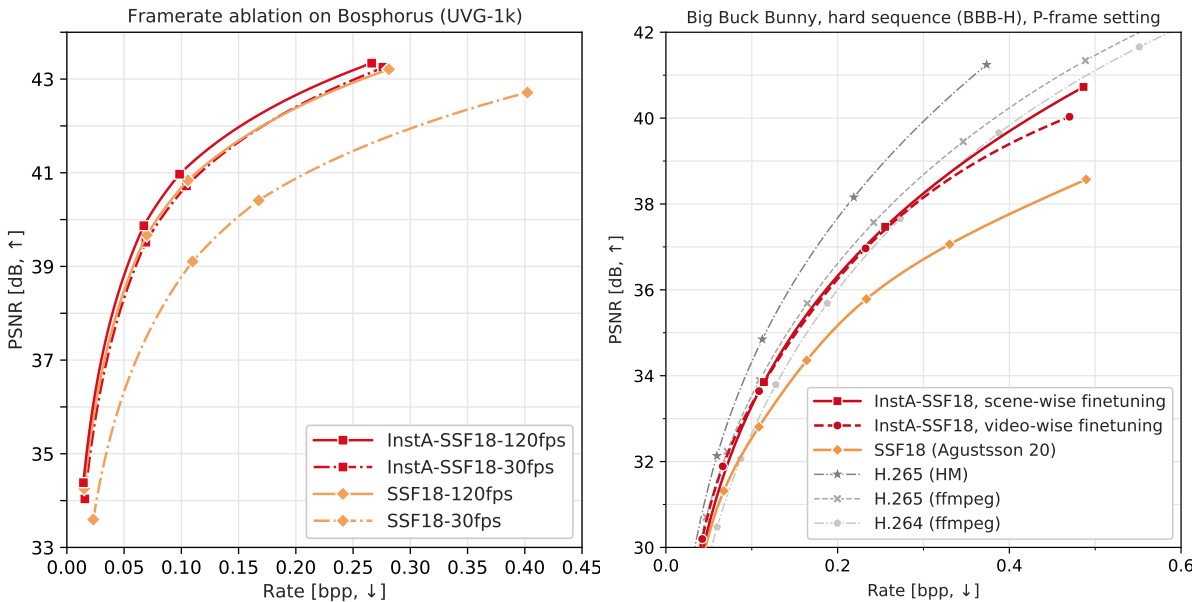

Figure 8: *Left:* Framerate ablation study. We compare the SSF18 baseline (orange) and InstA-SSF18 (red) on the original video at 120 fps (solid) and on the temporally subsampled video at 30 fps (dot-dashed). Instance-adaptive compression increases robustness to variation in framerate. *Right:* Full-video finetuning ablation study. We compare the performance of an SSF model finetuned on the full Big Buck Bunny video (10 minutes) to a SSF model finetuned only on the BBB-H scene (10 seconds) when evaluated on the BBB-H scene.

models. In general, we observe that instance-adaptive training improves the performance of models of any complexity. Instance adaption reduces the gap between smaller models and the full-size model, confirming that less expressivity is necessary for good performance on a single instance compared to on the entire data distribution.

## D.2   GoP size considerations

Throughout this work, we use a GoP size of 12 at test time. To test the robustness of this choice, in the right panel of Fig. 7 we compare to an infinite GoP size (for the SSF model). In other words, we compress the first frame as I-frame and all remaining frames as P-frames. The overall difference is slim, though the gap is larger for some videos than others.

## D.3   Sensitivity to framerate variations

To further quantify our robustness to test data characteristics, we investigate the effect of the video framerate on the model performance. We study the performance of global and instance-adaptive SSF models on temporally subsampled videos, i. e. keeping one out of every four frames. To save computational resources, this ablation is only performed on the "Bosphorus" sequence from the UVG-1k dataset. The left panel of Fig. 8 shows our results. While the SSF18 performance varies greatly between the original and the subsampled video, the difference in performance is much smaller for InstA-SSF18. In addition, both models perform better in the 120 fps scenario due to the higher redundancy across frames.

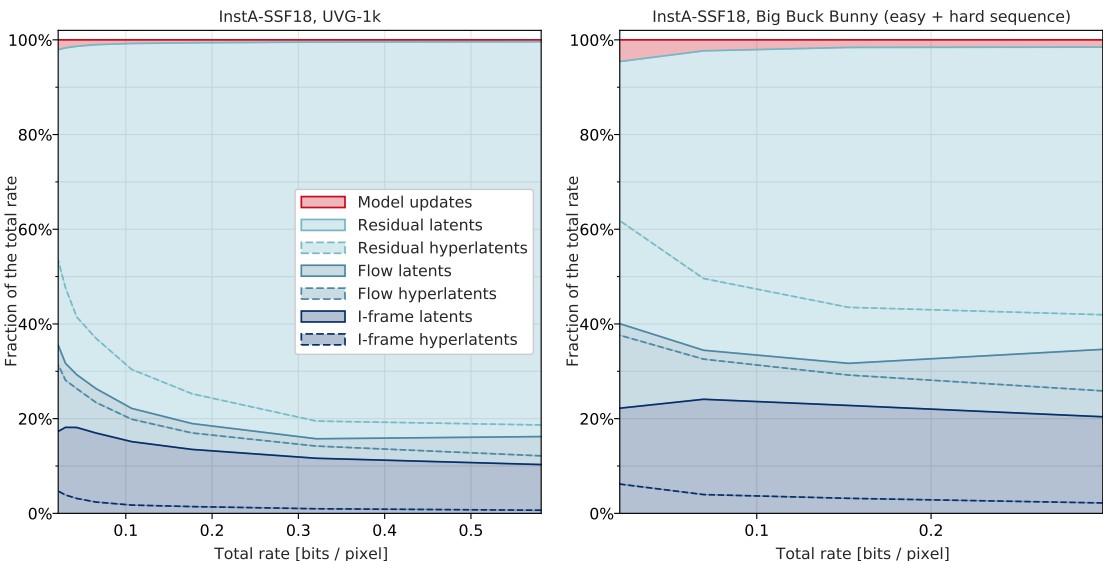

Figure 9: Bitrate allocation for our InstA-SSF18 model. We show the relative contribution to the overall bitrate from quantized latents for the I-frames, P-frame flow, and P-frame residuals (different shades of blue). In red we show the relative contribution from the quantized model updates. *Left*: on the UVG-1k dataset, which has similar content to the training data. *Right*: on the two Big Buck Bunny scenes, which differ substantially from the training data.

## D.4 Bitrate allocation

Figure 9 disentangles the various contributions to the total bitrate. We find that the overhead from model updates is generally small (typically well below 1 %), except on videos that differ substantially from the training data: on the animated Big Buck Bunny sequences, InstA learns to adapt more strongly and up to 3 % of the total rate are spent on model updates.

We further break down the model rate in Fig. 10, where we show how the model rate is distributed over the various auto-encoders and submodules of our InstA-SSF18 and InstA-SSF5 model. It can be seen that the InstA-SSF5 model has a lot fewer parameters per module by design, but it is making up for its smaller capacity by spending a lot more bits/per parameter on model updates compared to InstA-SSF18 (often more than double). This explains why the rate-distortion performance of InstA-SSF5 is close to that of InstA-SSF18. Even though the InstA-SSF5 model needs more bit per parameter to adapt to the data instances, the total bitrate spent on model updates is still smaller than that of InstA-SSF18.

Focusing on the distribution of the model bitrate across auto-encoders, we see that more than half of this rate is allocated to the I-frame model. The flow autoencoder is the next biggest consumer of model rate, while only few bits are spent to communicate updates to the residual autoencoder. In terms of submodules, it turns out that even though the hyper-decoder has a lot more parameters than the decoder, the decider receives substantially larger update bits per parameter. Together, these findings suggest that instance adaptation is most important for network components that are closer to the data space in the computational graph.

## D.5 Update sparsity analysis

The key design for achieving a low model rate is in update sparsity. We achieve this by using a spike-and-slab prior around global model parameter values. Training with the spike-and-slab prior is sparsity enforcing (on average around 0.8 % of all quantized parameters receive a non-zero update). To quantify savings of our chosen spike-and-slab prior, we compute the cross-entropy of our prior with the resulting model $\delta$ after training. This is the number of bits we need to transmit the model update. On the other hand, assume we had a Gaussian prior with the same variance as our "slab" component, and assuming the resulting $\delta$ follows

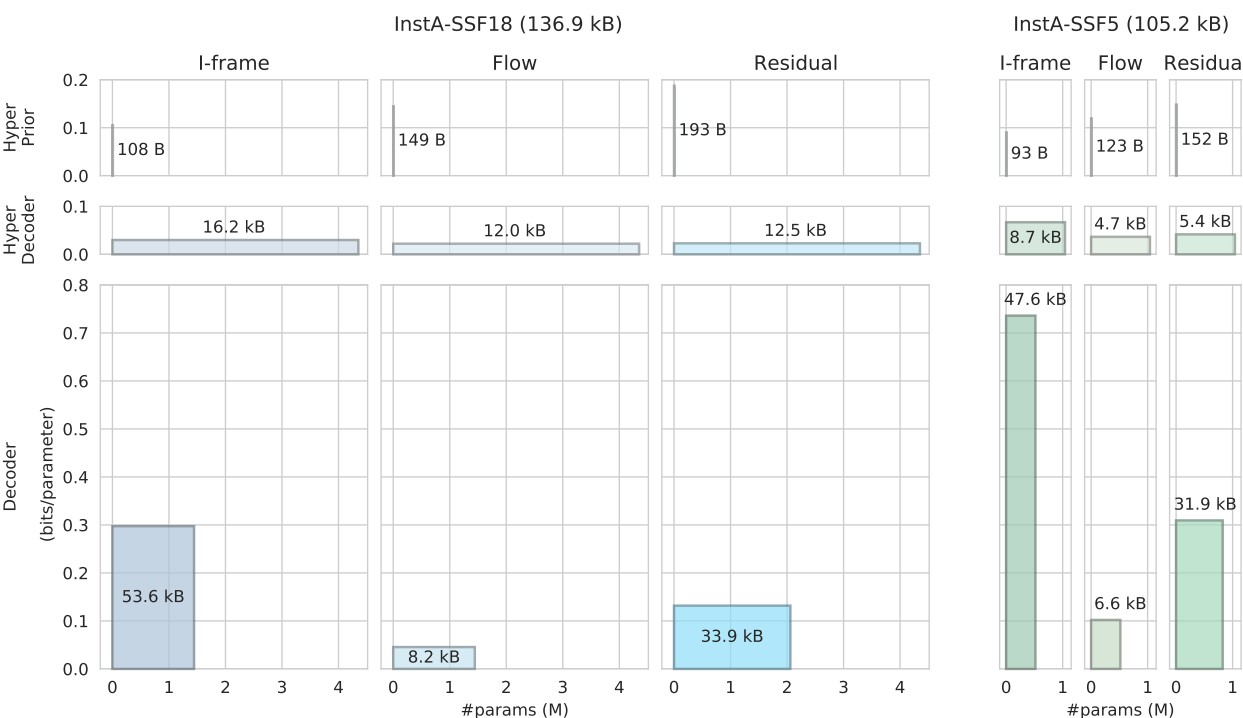

Figure 10: Model bitrate allocation for our InstA-SSF18 (left) and InstA-SSF5 (right) model. We show how the model rate is distributed across each autoencoder (columns) and submodule (rows). All plots use a grid of the same size, where width corresponds to the number of parameters in that module, height corresponds to the average bits/parameter used for updating that module, and area corresponds to total number of bits allocated for that module (each block in the grid has an area of 12.5 kB). Results are averaged over all values of $\beta$ and all datapoints in the UVG-1k and HEVC class B dataset.

the same distribution, the entropy of this Gaussian prior gives the expected number of bits to transmit non-sparse model update. We estimate the difference between the two scenarios to be around 7.5 bits per parameter.

### D.6 Finetuning strategy for longer videos

For longer videos with multiple scenes, one can consider several different strategies for instance-adaptive finetuning. The simplest approach is finetuning a single model on the full video. It may be beneficial to instead finetune the model independently on each scene. Even a hierarchical approach in which a model finetuned on the full video is further finetuned on each scene is an option.

The optimal strategy will in general depend on the particular video content and the similarity of the scenes as well as on the bitrate setting. We perform a simple first test by finetuning a single model on the entire Big Buck Bunny video, a 10 min video (14315 frames) with multiple scenes united by a common style. The results in the right panel of Fig. 8 show that such a single finetuned model leads to clear RD improvements over the global SSF model. However, in the large-bitrate region these improvements are smaller than the gains we achieve by finetuning on the BBB-E and BBB-H scenes individually.

## E  Broader Impact Statement

Video compression affects the life of millions of people and any algorithm and model should be carefully analyzed for potential ethical issues before deployment. In particular, it is important to check for harmful biases, which may be due to imbalanced training data, but can also be caused or exacerbated by algorithmic choices. To some extent, the instance-adaptive video compression method we are proposing may be helpful in combating such biases, as it can improve the compression performance on data that is different from typical samples in the training set. However, it cannot solve the problem entirely, as more pronounced instance adaptation will increase the bitrate.

The environmental impact of our research is also relevant. The energy cost of our experiments directly contributes to the climate catastrophe. In the long term, our work may also have a beneficial aspect, as it allows us to lower the bitrate and thus the energy cost of video streaming, which already now make up the majority of internet traffic.

## F  Reproducibility statement

Our instance-adaptive video compression algorithm is described in detail in Sec. 3. The most complex components are the base compression models, which are described in more detail in the references we cite as well as in appendix B. With the details we provide, instance-adaptive video compression can be implemented and used for any base model with relatively little effort. Finally, in the supplementary material we provide the performance of the various methods on each video sequence as a CSV file to aid comparisons.

