# OpenReview forum: "Instance-Adaptive Video Compression: Improving Neural Codecs by Training on the Test Set"
_TMLR — Accepted by TMLR_

### Review · Reviewer_J84y · 2023-03-30

**Summary Of Contributions:**

The paper proposes a method to improve the bitrate-distortion performance of learning-based video compression algorithms using instance-adaptive learning. The parameters of the variational auto-encoder for compression are finetuned on the specific video sequence to compress and the parameter updates of the decoder are transmitted in an efficient way along the compressed video. The method has been evaluated on various datasets and in comparison to reasonable baselines showing significant BD-rate savings.

**Audience:**

Yes

**Claims And Evidence:**

Yes

**Requested Changes:**

- Some clarifications as metioned in weaknesses.
- Explaining/discussion the expensive finetuning time.
- Typo: Caption Figure 8: emphLeft -> \emph{Left} ?
- Minor: Adding the used GPU in the caption of Figure 5.
- Given that the encoding time is one of the main concerns, it would be interesting to add the analysis of the improvement wrt. to the encoding also for the B-frame setting (analog to figure 5).

**Strengths And Weaknesses:**

#### Strengths
- The paper is well-written.
- The evaluation has been performed in the B-frame and P-frame settings, showing that the approach is working in both settings.
- The improvement is not equally large in all settings. But limitations, such as that the model cannot compete with traditional methods in case of difficult out-of-distribution examples have been discussed. It still performs better than the learning-based baselines.

#### Weaknesses
The paper could benefit from some clarifications:
- Abstract claims that the proposed algorithm is model agnostic. However, the method section assumes a VAE structure. This could be more precisely formulated.
- The method builds upon the image-based approach van Rozendaal et al. (2021). Section 3.4 could be clearer about what is part of the related paper and which modifications were needed to adapt it to the video setting to make the technical contributions clearer.

Because the method is instance-adaptive, it requires finetuning for each instance. Section 4.3 mentions that this finetuning is performed for two weeks for the main results, which is very expensive to encode a single 2.5-10s sequence. This makes me wonder about the practicality of the method.

---

> ### Author Response · Authors · 2023-04-28
> **Response to Reviewer J84y**
>
> We thank the reviewer for their comments and suggestions.
>
> We acknowledge that we only test the method on compressive autoencoders, as these are the de-facto standard in neural compression.
> In theory, the proposed model prior and optimization method are agnostic to the model architecture and could be applied to many other compression networks, from GAN-based codecs to methods based on neural implicit representations.
>
> We make the following clarifications:
>   1. We have fixed the typo in the Figure caption and thank the reviewer for spotting it.
>   2. We clarified our contribution with respect to van Rozendaal et al in Section 3.4,
>   3. We have added the GPU type to the caption of Figure 5.
>
> > Because the method is instance-adaptive, it requires finetuning for each instance. Section 4.3 mentions that this finetuning is performed for two weeks for the main results, which is very expensive to encode a single 2.5-10s sequence. This makes me wonder about the practicality of the method.
>
> The reviewer rightly points out that finetuning in the instance-adaptive setting can be very expensive.
> However, we emphasize that most of the RD gains are achieved within the first 1-20 seconds/frame of encoding (training for 8 minutes - 1.5 hours), as can be seen in Figure 5.
> Additionally, although slow encoding is not useful for all applications, it is very valuable for the use case where a video is encoded once, and transmitted and decoded many times. This is the case for most on-demand video and video streaming services.
> This slow-encoding use case has been covered in related work, as can be seen in Table 1. The strongest standard codecs typically allow slow encoding as well in order to improve compression performance as can be seen in Figure 5. For example, the encoding time of ffmpeg can be configured by the user.
> These considerations are discussed in our paper in Sections 5.7 and 6.
>
> > Given that the encoding time is one of the main concerns, it would be interesting to add the analysis of the improvement wrt. to the encoding also for the B-frame setting (analog to figure 5).
>
> We do agree with the reviewer that this would be interesting to visualize.  We chose to not include this plot as it requires additional experiments and evaluations (for both standard and neural codecs). Based on the loss curves observed during training, we have no reason to believe the qualitative behavior is any different than that of the SSF model.

---

### Review · Reviewer_DBHN · 2023-04-09

**Summary Of Contributions:**

This paper introduces an instance adaptive video compression method, by finetuning a pretrained neural compression model on the video sequence to be transmitted. The finetuned network parameters are compressed and transmitted along with the latent code under the spike-slab update prior. The proposed method is proved to be able to improve the performance and can also reduce the requirements of the model parameters and MACs.

**Audience:**

Yes

**Broader Impact Concerns:**

No.

**Claims And Evidence:**

Yes

**Requested Changes:**

1. The paper writing should be improved. There are many language organization confusion and typographical problems.
2. The introduction part is too simple, which should explain clearly the motivation of this work. In addition, the introduction of the concept of instance learning is also missing.
3. There are many experiments in this article, but a comparison with similar "fine-tuning" methods is missing to prove the superiority of the proposed one.


**Strengths And Weaknesses:**

Strengths
1. The proposed method is well understood and the code is easily reproducible.
2. The performance improvement is significant and the property that the number of neural network model parameters can be saved is valuable.

Weaknesses
1. This work is migrated from a similar work on image compression, lacking some thought about the optimization of video compression methods.
2. Fine-tuning the model on the test set is a relatively naive approach at this moment, especially with the advent of implicit neural networks. The introduction of the finetuning time on test set, and extra complexity of both encoder and decoder side makes the method look less friendly.

---

> ### Author Response · Authors · 2023-04-28
> **Response to Reviewer DBHN**
>
> We thank the reviewer for their comments.
>
> The reviewer points out that a comparison to similar finetuning methods is missing. We do perform encoder-side finetuning (Lu et al., 2020) - which, for a large enough encoder, is as strong as direct latent optimization - as a baseline.
>
> We made the following changes in response:
>   1. We clarified our contribution with respect to van Rozendaal et al in Section 3.4.
>   2. We have included a more elaborate discussion about the advantages and downsides of neural implicit models in the related work section.
>
> > The paper writing should be improved. There are many language organization confusion and typographical problems.
>
> Thank you for the feedback. Could you point us to any concrete examples so that we can improve the paper?
>
> > The introduction part is too simple, which should explain clearly the motivation of this work.
>
> We thank the reviewer for their suggestion and have reorganized the introduction of our paper to better emphasize the motivation.
>
> > In addition, the introduction of the concept of instance learning is also missing.
>
> The term instance-adaptive compression is introduced in the penultimate paragraph of the introduction. We hope the reorganized introduction will address this concern.
>
> > There are many experiments in this article, but a comparison with similar "fine-tuning" methods is missing to prove the superiority of the proposed one.
>
> We thank the reviewer for their comment. We would like to point out that we already included a well-tuned baseline for encoder-side finetuning. This is the most simple approach to overfitting as it does not require any changes on the decoder side. We show in Figures 1 and 2, and Table 3, 4 and 5 that our method improves significantly over this baseline.
>
> > This work is migrated from a similar work on image compression, lacking some thought about the optimization of video compression methods.
>
> We have updated our manuscript to further clarify our contribution with respect to van Rozendaal et al., 2021. Our main contribution is the experimental evaluation of the method on video sequences. We have benchmarked our Instance-Adaptive finetuning scheme on different models in the P-frame and B-frame settings on multiple video datasets. In addition, we included video-specific ablations examining the effect of model size (Figure 8, left), GoP size (Figure 8, right), framerate (Figure 9 left), and video length (Figure 9, right).
>
> > Fine-tuning the model on the test set is a relatively naive approach at this moment, especially with the advent of implicit neural networks. The introduction of the finetuning time on test set, and extra complexity of both encoder and decoder side makes the method look less friendly.
>
> Indeed, neural implicit models form a very important class of related work.
> We have included them in our related work section as well as our experiments, and show that our InstA-B-EPIC model greatly outperforms the NeRV model (Chen et al., 2021).
>
> Based on the reviewer's question we have extended the related work section to make a more detailed comparison between neural implicit methods and our finetuning-based approach.
>
> While neural implicit methods have some advantages over our method, such as being easier to standardize and less susceptible to data bias, they also come with disadvantages. The biggest issue is that for most neural implicit methods, the entire bitstream needs to be received before the first frame can be decoded. This makes them impractical for longer videos and for streaming settings.  In contrast, our method only requires a very small fraction of the bitstream for the model updates, after which the video can be decoded following normal VAE protocol.
> Furthermore, our method benefits from the fact that even after finetuning for 0 steps, the method is already as good as the baseline.
> Lastly, the current implicit compression models require a 4-stage pipeline of training followed by pruning, quantization and Huffman coding. Instead, our models are trainable end-to-end (using quantization and a rate loss during training) and only require entropy coding after the model is overfitted.

---

### Review · Reviewer_6RHk · 2023-04-24

**Summary Of Contributions:**

This paper proposes a new instance-adaptive video compression method. The basic assumption of this method is that there could be domain gap between the training data and the test dat such that the model that is well fitted on the training data does not generalize well on the test video. The key contribution is to finetune the codec for each video instance. Then both the latent code of the video, and the parameter updates of the decoder and prior are transmitted. To reduce the bitrates for the parameter updates, a spike-and-slab prior is assumed for the updates. In addition, the updates are also quantized. In the decoder side, the updates of the parameters are first decoded. Then the decoder and the prior network are updated accordingly. After that, the video is decoded with the updated network.

**Audience:**

Yes

**Broader Impact Concerns:**

The authors discussed two issues about the broader impact. The first one is to reduce the biases due to insufficient training data. The second one is the energy reduction due to the reduced bitrate. Yet, I'm interested in the cost to finetune the models for each instance. Is that overhead acceptable?

**Claims And Evidence:**

Yes

**Requested Changes:**

1. The figures and tables should be placed near the text in the paper. In the current version, it is difficult to relate the figures and the explanation.
2. The bitrate and decoding overhead should be fully analyzed. In Table 2, there are 4 versions of SSF.
3. Figure 2 should be better explained. For example, why are some of the curves not monotomous?  What does percentage on the y-axis mean? Why are most of the curves descreasing? Should a higher relative rate correspond to a higher PSNR?
4. What is the distribution of the updates? It is better to visualize the distribution.
5. How is the spike-and-slab prior enforced? Is it used as a regularization term?
6. What is computational cost to finetune the codecs?
7. Although this work is about data compression, network compression is closely related. For example, the network updates should be quantized. Thus, there are two questions regarding that.
1) In the related works, network compression techniques should be discussed a little bit [1-3].
2) Is it possible to compress the network during the finetuning of them to achieve a reduction of both parameters and computational cost [1-3]?

[1] Li, Y., Dong, X., & Wang, W. (2019). Additive powers-of-two quantization: An efficient non-uniform discretization for neural networks. arXiv preprint arXiv:1909.13144.

[2] Li, Y., Gu, S., Mayer, C., Gool, L. V., & Timofte, R. (2020). Group sparsity: The hinge between filter pruning and decomposition for network compression. In Proceedings of the IEEE/CVF conference on computer vision and pattern recognition (pp. 8018-8027).

[3] Liu, Z., Mu, H., Zhang, X., Guo, Z., Yang, X., Cheng, K. T., & Sun, J. (2019). Metapruning: Meta learning for automatic neural network channel pruning. In Proceedings of the IEEE/CVF international conference on computer vision (pp. 3296-3305).

**Strengths And Weaknesses:**

Strengths:
1. Compared with the previous methods, the proposed method achieves quite a decent improvement of the rate-distortion performance.
2. The method is model-agnostic and could be applied to any codecs.

Weakness:
1. The writing could be improved. (See requested changes)
2. The novelty in the method is incremental. Most of the part is based on SSF18.

---

> ### Author Response · Authors · 2023-04-28
> **Response to Reviewer 6RHk**
>
> We thank the reviewer for their comments and suggestions.
>
> We implemented the following changes:
>   1. We improved the placement of figures so that they better match the location in the text.
>   2. We added an explanation for the interpolation procedure in Figure 2. This plot does not show the absolute bitrate, but the bitrate overhead relative to another compression baseline. This means curves do not have to be monotonically increasing.
>   3. In Table 2, where we show the four different SSF variants, we now refer to Figure 8, where we benchmark them. For conciseness, we mainly focus on SSF18 and SSF5 throughout the rest of the paper.
>   4. We have included a paragraph about model compression methods in the related work section and refer to all of the requested references.
>
> Indeed, the novelty is not in the architecture. Instead, we demonstrate that the instance-adaptive optimization method works for multiple architectures and that it allows using smaller network architectures that do not need to generalize. For TMLR, we believe substantial novelty is not expected and our elaborate experimental evaluations and strong rate-distortion improvements provide valuable insights for the compression community.
>
>
> In the following, we would like to answer the reviewer's questions.
>
> > Figure 2 should be better explained. For example, why are some of the curves not monotomous? What does percentage on the y-axis mean? Why are most of the curves descreasing? Should a higher relative rate correspond to a higher PSNR?
>
> This way of plotting can be found in other data compression papers such as Yang et al., (2020) and Zhu et al. (2022). Intuitively, it shows "how many bits would we use relative to a baseline (the line at 0%) at a given PSNR", where a negative number indicates fewer bits are needed. The y-axis shows the relative rate-savings compared to a baseline for the interpolated R/D curves. The x-axis shows the PSNR.  Although relative scores may be difficult to parse at first glance, they provide a comparison that is hard to see in the rate-distortion plot of Figure 1, especially for the lower (33 to 36) PSNR range. The widely used Bjontegaard Delta-rate metric is based on a similar idea: it is calculated by integrating the relative rate (shown on the Y-axis here) for a certain PSNR range (shown on the X-axis here).
>
> > What is the distribution of the updates? It is better to visualize the distribution.
>
> The model updates have a very sparse distribution, on average around 90 \% of all quantized parameters receive a zero update. Because the distribution is so sparse we decide to not show any histograms as they would only show a delta peak.
>
> > How is the spike-and-slab prior enforced? Is it used as a regularization term?
>
> Correct. We overfit end-to-end on the loss from equation 2, which includes a regularization term that enforces the spike-and-slab prior.
>
> > What is computational cost to finetune the codecs?
>
> As shown in Figure 5 and discussed in section 5.7 in the paper, most of the benefits from instance-adaptive finetuning materialize within a few seconds of finetuning time per frame. Finetuning for longer (we tested this until two weeks per video sequence) leads to further improvements. The optimal trade-off of encoding compute and compression performance will depend on the use-case: fine-tuning for a substantial amount of time may be useful in one-to-many scenarios, for example when a single video is sent thousands of times.
>
> > Is it possible to compress the network during the finetuning of them to achieve a reduction of both parameters and computational cost [1-3]?
>
> Since we compress and transmit network parameter updates and not the network parameters themselves, this is not trivial; a sparse update does not imply a sparse parameter. However, we do show that the finetuning approach allows us to decrease the network size by 72% without sacrificing compression performance. This leads to a large reduction in complexity both during encoding (overfitting) and decoding of the video.
>
>
> 1. Yang, Yibo, Robert Bamler, and Stephan Mandt. "Improving inference for neural image compression." Advances in Neural Information Processing Systems 33 (2020): 573-584.
> 2. Zhu, Yinhao, Yang Yang, and Taco Cohen. "Transformer-based transform coding." International Conference on Learning Representations. 2022.

---

### Decision · Action_Editors · 2023-06-09

**Recommendation:** Accept as is

**Comment:**

Three expert reviewers review this paper. The discussions with the authors clarify most of the initial concerns (as outlined by the authors in the "Changes Since Last Submission" section). The revised manuscript has been significantly improved. All three reviewers are satisfied with the revision and recommend learning accept. The AE reads the reviews, the discussions, and agrees with the reviewers to recommend accepting the paper.

**Audience:**

Yes, I believe video compression is an important topic for TMLR's audience.

**Claims And Evidence:**

The paper presents an instance-adaptive video compression method. The experiments validate the contributions via rate-distortion performance improvement over the state-of-the-art and its applicability to several codecs and P-frame and B-frame settings.